# Equivariant Eikonal Neural Networks: Grid-Free, Scalable Travel-Time Prediction on Homogeneous Spaces

**Alejandro García-Castellanos**[1][*]     **David R. Wessels**[1,2]     **Nicky J. van den Berg**[3]
**Remco Duits**[3]     **Daniël M. Pelt**[4]     **Erik J. Bekkers**[1]
[1]Amsterdam Machine Learning Lab (AMLab), University of Amsterdam     [2]New Theory
[3]Department of Mathematics and Computer Science, Eindhoven University of Technology
[4]Leiden Institute of Advanced Computer Science, Universiteit Leiden

## Abstract

We introduce Equivariant Neural Eikonal Solvers, a novel framework that integrates Equivariant Neural Fields (ENFs) with Neural Eikonal Solvers. Our approach employs a single neural field where a unified shared backbone is conditioned on signal-specific latent variables – represented as point clouds in a Lie group – to model diverse Eikonal solutions. The ENF integration ensures equivariant mapping from these latent representations to the solution field, delivering three key benefits: enhanced representation efficiency through weight-sharing, robust geometric grounding, and solution steerability. This steerability allows transformations applied to the latent point cloud to induce predictable, geometrically meaningful modifications in the resulting Eikonal solution. By coupling these steerable representations with Physics-Informed Neural Networks (PINNs), our framework accurately models Eikonal travel-time solutions while generalizing to arbitrary Riemannian manifolds with regular group actions. This includes homogeneous spaces such as Euclidean, position–orientation, spherical, and hyperbolic manifolds. We validate our approach through applications in seismic travel-time modeling of 2D, 3D, and spherical benchmark datasets. Experimental results demonstrate superior performance, scalability, adaptability, and user controllability compared to existing Neural Operator-based Eikonal solver methods.

## 1   Introduction

The eikonal equation is a first-order nonlinear partial differential equation (PDE) that plays a central role in a wide range of scientific and engineering applications. Serving as the high-frequency approximation to the wave equation [Noack and Clark, 2017], its solution represents the shortest arrival time from a source point to any receiver point within a specified scalar velocity field [Sethian, 1996]. This formulation underpins numerous applications: in Computer Vision, it is integral to the computation of Signed Distance Functions (SDFs) [Jones et al., 2006] and geodesic-based image segmentation [Chen and Cohen, 2019]; in Robotics, it facilitates optimal motion planning and inverse kinematics [Ni and Qureshi, 2023, Li et al., 2024b]; and in Geophysics, it models seismic wave propagation through heterogeneous media, enabling critical travel-time estimations [Abgrall and Benamou, 1999, Rawlinson et al., 2010, Schuster and Quintus-Bosz, 1993].

Conventional numerical solvers, such as the Fast Marching Method (FMM) [Sethian, 1996] and the Fast Sweeping Method (FSM) [Zhao, 2004], have historically been used to compute solutions to the eikonal equation. However, these approaches are heavily dependent on spatial discretization, leading

---

[*]Corresponding author: <a.garciacastellanos@uva.nl>

39th Conference on Neural Information Processing Systems (NeurIPS 2025).

to a challenging trade-off: higher resolution is required for complex velocity models, which in turn dramatically increases computational and memory demands [Grubas et al., 2023, Song et al., 2024, Mei et al., 2024, Smith et al., 2021, Waheed et al., 2021]. This issue is exacerbated in scenarios involving complex input geometries, such as Riemannian manifolds, which are prevalent in both computer vision [Bekkers et al., 2015] and robotics applications [Li et al., 2024b].

Recent advances in scientific machine learning have introduced neural network-based solvers as promising alternatives. Physics-Informed Neural Networks (PINNs) integrate the PDE constraints into the training loss, offering a grid-free approximation to the eikonal equation and alleviating the discretization issues inherent in traditional numerical methods [Smith et al., 2021, Waheed et al., 2021, Ni and Qureshi, 2023, Grubas et al., 2023, Li et al., 2024b]. However, a significant limitation of PINN-like approaches is their requirement to train a new network for each distinct velocity field, which hampers their applicability in real-time scenarios.

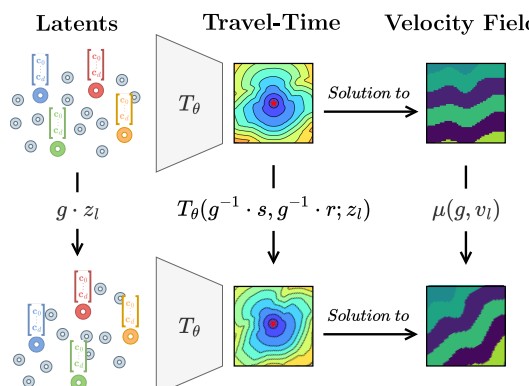

Figure 1: Steerability in the Equivariant Neural Eikonal Solver (E-NES) enables weight sharing across the entire group orbit: applying a group transformation to the conditioning variable $z_l$, induces a corresponding transformation on the travel-time function $T_\theta(\cdot, \cdot; z_l)$ through the group's left regular representation, and on the associated velocity field $v_l$ via the non-linear group action $\mu$, as formalized in Section 4.1.

Neural Operators overcome this constraint by learning mappings between function spaces – specifically, from velocity fields to their corresponding travel-time solutions. Unlike PINNs, neural operators utilize a shared backbone and incorporate conditioning variables to handle different velocity profiles [Song et al., 2024, Mei et al., 2024]. Current approaches leveraging simple architectures such as DeepONet [Mei et al., 2024] and Fourier Neural Operators [Song et al., 2024] have demonstrated promising results, yet there remain several avenues for improvement, as discussed in Section 2.

As stated in Wang et al. [2024b], one promising direction is to recognize that Neural Operators belong to a broader class of models known as Conditional Neural Fields. These models, which have been popularized within the Computer Vision community, explore advanced conditioning techniques to enhance expressivity, adaptability, and controllability [Dupont et al., 2022, Wessels et al., 2024, Wang et al., 2024b]. In this work, we focus on the recently introduced Equivariant Neural Fields, which ground these conditioning variables in geometric principles, leading to improved representation quality and steerability – ensuring that transformations in the latent space correspond directly to transformations in the solution space [Knigge et al., 2024].

Our key contributions are as follows:

- We introduce a novel, expressive generalization of Equivariant Neural Fields to functions defined over products of Riemannian manifolds with regular group actions, including homogeneous spaces associated with linear Lie groups.
- We implement this framework through our Equivariant Neural Eikonal Solver (E-NES), to efficiently solve eikonal equations by leveraging geometric symmetries, enabling generalization across group transformations without explicit data augmentation (see Figure 1).
- We validate our approach through comprehensive experiments on 2D, 3D, and spherical seismic travel-time benchmarks, achieving superior scalability, adaptability, and user controllability compared to existing methods in a grid-free manner.

Our code, including scripts to generate the results, is provided at: https://github.com/AGarciaCast/E-NES.

## 2 Related work

**Neural eikonal solvers.** Initial neural approaches for solving the eikonal equation have primarily relied on Physics-Informed Neural Networks (PINNs), which incorporate the PDE directly into

the loss function [Smith et al., 2021, Waheed et al., 2021, Ni and Qureshi, 2023, Grubas et al., 2023, Li et al., 2024b, Kelshaw and Magri, 2024]. These models are trained individually for each velocity field, achieving high-accuracy reconstructions at the expense of significant computational and memory overhead. Moreover, this per-instance training lacks cross-instance generalization, limiting its practicality for large-scale or real-time applications.

To enable generalization across velocity fields, operator learning methods have been proposed. DeepONet variants [Lu et al., 2021, Mei et al., 2024] learn mappings between function spaces, but typically require discretization of either the source or receiver points. This limits resolution invariance and complicates applications requiring continuous evaluations, such as geodesic path planning. Moreover, some methods, such as Mei et al. [2024], rely on the supervision of a numerical solver, which can be beneficial in some simple scenarios but is a bottleneck in complex ones. On the hand, other operator learning approaches such as Fourier Neural Operators [Song et al., 2024] offer solver-free alternatives and incorporate the PDE into the loss, but still rely on partial discretization, inheriting similar resolution constraints.

In contrast, our method avoids discretization entirely by representing both inputs continuously and training solely with PDE supervision. We adopt the Conditional Neural Field (CNF) framework as our backbone, enabling scalable conditioning on velocity fields while preserving grid-free inference and resolution independence.

**Conditional and equivariant neural fields.** Conditional Neural Fields (CNFs) are coordinate-based networks that reconstruct continuous signals from discrete observations. Formally, a CNF $f_\theta : \mathcal{M} \times \mathcal{Z} \to \mathbb{R}^d$ maps coordinates $p \in \mathcal{M}$ and latent codes $z \in \mathcal{Z}$ to outputs $f_\theta(p; z)$ approximating target signals. Given a dataset $\mathcal{D} = \{f_i\}_{i=1}^n$, a single network can represent all signals via instance-specific latents: $f_\theta(p; z_i) \approx f_i(p)$.

Early CNFs used global latent vectors for conditioning [Dupont et al., 2022]. Subsequent work demonstrated that learnable point cloud latents $\{z_i\}_{i=1}^m \subseteq \mathcal{Z}$ significantly enhance expressivity and reconstruction quality [Bauer et al., 2023, Luijmes et al., 2025, Kazerouni et al., 2025, Wessels et al., 2024]. Equivariant Neural Fields (ENFs) further impose symmetry priors through the *steerability property*: $f_\theta(g^{-1} \cdot p; \{z_i\}) = f_\theta(p; \{g \cdot z_i\})$ for all $g \in G$. This equivariant design improves sample efficiency and generalization [Wessels et al., 2024, Chen et al., 2022, Chatzipantazis et al., 2023] by encoding geometric structure directly into the latent space.

In this work, we *extend ENFs to signals defined on products of Riemannian manifolds*, i.e., $f : \mathcal{M}_1 \times \cdots \times \mathcal{M}_n \to \mathbb{R}^d$, enabling complex inter-input interactions while preserving equivariance. This generalization substantially broadens the scope of addressable problems beyond prior single-point formulations, with implications extending beyond eikonal solving (see Appendix F).

**Invariant function learning.** Steerability in ENFs requires invariance to joint transformations: $f(g \cdot p; \{g \cdot z_i\}) = f(p; \{z_i\})$ for all $g \in G$ [Wessels et al., 2024, Chen et al., 2022, Chatzipantazis et al., 2023]. Constructing expressive invariant functions is thus central to our framework.

We provide a formal expressivity analysis establishing a *complete and maximally expressive* set of independent invariants, guaranteeing zero information loss during canonicalization. This addresses a critical gap in prior work [Wessels et al., 2024, Knigge et al., 2024], where invariants were selected heuristically without completeness guarantees. Our analysis extends to arbitrary (possibly non-transitive) Lie group actions on product manifolds.

From Invariant Theory, a complete set of *fundamental invariants* must: (1) express any invariant function, and (2) separate orbits—i.e., $I_\nu(p) = I_\nu(q)$ for all invariants $I_\nu$ if and only if $p$ and $q$ share the same orbit [Olver, 1995]. While several computational approaches exist—including Weyl's theorem [Weyl, 1946, Villar et al., 2021], infinitesimal methods [Andreassen, 2020], moving frames [Olver, 2001], and differential invariants [Olver, 1995, Sangalli et al., 2022, Li et al., 2024a]—we employ *the moving frame technique* [Olver, 2001] for its conceptual clarity and natural connection to modern canonicalization methods [Shumaylov et al., 2024] (see Appendix B for detailed comparisons).

Moreover, rather than relying on global canonicalization—which produces a single canonical representation for an entire point cloud—we adopt a local canonicalization strategy [Hu et al., 2024, Wessels et al., 2024, Chen et al., 2022, Du et al., 2022, Wang et al., 2024a, Zhang et al., 2019]. By canonicalizing small patches, our approach is better able to capture relevant local information and

facilitates the use of transformer-based architectures, as opposed to the DeepSet-based architectures commonly employed in Blum-Smith et al. [2024], Dym and Gortler [2023], Villar et al. [2021].

## 3 Background

In this section, we present the necessary mathematical foundations and formulate the eikonal equation problem. For a comprehensive treatment of these topics, we refer the reader to Lee [2018], Olver [1995].

### 3.1 Mathematical Preliminaries

**Differential Geometry.** A *Riemannian manifold* is defined as a pair $(\mathcal{M}, \mathcal{G})$, where $\mathcal{M}$ is a smooth manifold and $\mathcal{G}$ is a Riemannian metric tensor field on $\mathcal{M}$ [Lee, 2018]. The metric tensor $\mathcal{G}_p : T_p\mathcal{M} \times T_p\mathcal{M} \to \mathbb{R}$ assigns to each $p \in \mathcal{M}$ a positive-definite inner product on the tangent space $T_p\mathcal{M}$. Specifically, for any tangent vectors $\dot{p}_1, \dot{p}_2 \in T_p\mathcal{M}$, the inner product is given by $\mathcal{G}_p(\dot{p}_1, \dot{p}_2)$, and the corresponding norm is defined as $\|\dot{p}_1\|_{\mathcal{G}} = \sqrt{\mathcal{G}_p(\dot{p}_1, \dot{p}_1)}$.

The *Riemannian distance* between two points $p$ and $q$ in a connected manifold $\mathcal{M}$, denoted as $d_{\mathcal{G}}(p, q)$, is defined as the infimum of the length of all smooth curves joining them [Lee, 2018]. Curves that achieve this infimum while traveling at constant speed are known as *geodesics*.

For a smooth scalar field $f : \mathcal{M} \to \mathbb{R}$, the *Riemannian gradient* $\operatorname{grad} f$ is the unique vector field reciprocal to the differential $\mathrm{d}f : T\mathcal{M} \to \mathbb{R}$, meaning $\mathcal{G}(\operatorname{grad} f, \cdot) = \mathrm{d}f$. Given a smooth function $\phi : \mathcal{M} \to \mathcal{M}$, we can also define the *adjoint of a differential* $\mathrm{d}\phi(p) : T_p\mathcal{M} \to T_{\phi(p)}\mathcal{M}$ at a point $p \in \mathcal{M}$ as the map $(\mathrm{d}\phi(p))^* : T_{\phi(p)}\mathcal{M} \to T_p\mathcal{M}$; such that for every $\dot{p} \in T_p\mathcal{M}, \dot{q} \in T_{\phi(p)}\mathcal{M}$ we have that $\mathcal{G}_{\phi(p)}(\mathrm{d}\phi(p)[\dot{p}], \dot{q}) = \mathcal{G}_p(\dot{p}, (\mathrm{d}\phi(p))^*[\dot{q}])$ [Lezcano-Casado, 2019].

Note that when $\mathcal{M} = \mathbb{R}^n$ and $\mathcal{G}_p = \mathbf{I}_n$ for all $p \in \mathbb{R}^n$, all Riemannian notions reduce to their Euclidean counterparts.

**Group Theory.** A *Lie group* $G$ is a smooth manifold with group operations that are smooth. A (left) *group action* on a set $X$ is a map $\mu : G \times X \to X$ satisfying $\mu(e, x) = x$ and $\mu(g, \mu(h, x)) = \mu(gh, x)$ for all $x \in X$, $g, h \in G$. When $\mu$ is clear by the context we write $g \cdot x$. The *orbit space* $X/G$ is the quotient space obtained by identifying points in $X$ that lie in the same orbit under the $G$-action. Formally, $X/G = \{\operatorname{Orb}(x) \mid x \in X\}$ consists of all distinct orbits, where each orbit $\operatorname{Orb}(x) = \{g \cdot x \mid g \in G\}$ represents an equivalence class under the relation $x \sim y \Leftrightarrow \exists g \in G$ such that $y = g \cdot x$. These equivalence classes partition $X$ into mutually disjoint subsets whose union equals $X$, yielding a canonical decomposition that reflects the underlying symmetry of the group action.

In this work, we will focus on the Special Euclidean group $SO(n) = \{R \in \mathbb{R}^{n \times n} \mid RR^T = \mathbf{I}_n, \ \det(R) = 1\}$ and the Special Euclidean group $SE(n) = \mathbb{R}^n \rtimes SO(n)$, representing roto-translations. For $g = (\mathbf{t}, R) \in SE(n)$, the group product is $g \cdot g' = (\mathbf{t}, R)(\mathbf{t}', R') = (\mathbf{t} + R\mathbf{t}', RR')$.

The *isotropy subgroup* (or stabilizer) at $x \in X$ is $G_x = \{g \in G \mid g \cdot x = x\}$. A group $G$ acts *freely* if $G_x = \{e\}$ for all $x \in X$, meaning no non-identity element fixes any point. An $r$-dimensional Lie group acts freely on a manifold $\mathcal{M}$ if and only if its orbits have dimension $r$ [Olver, 1995]. A group acts *regularly* on $\mathcal{M}$ if each point has arbitrarily small neighborhoods whose intersections with each orbit are connected. In practical applications, the groups of interest typically act regularly.

As discussed in Section 2, it is crucial for building Equivariant Neural Fields to obtain a complete set of functionally independent invariants. As demonstrated by Olver [2001], when a Lie group acts freely and regularly, such a set can be systematically derived locally using the moving frame method – detailed in the Appendix (Section B).

### 3.2 Eikonal Equation Formulation

On a Riemannian manifold $(\mathcal{M}, \mathcal{G})$, the two-point Riemannian *Eikonal equation* with respect to a velocity field $v : \mathcal{M} \to [v_{\min}, v_{\max}]$ (where $0 < v_{\min} \leq v_{\max} < \infty$) is:

$$\begin{cases} \|\operatorname{grad}_s T(s, r)\|_{\mathcal{G}} = v(s)^{-1}, \\ \|\operatorname{grad}_r T(s, r)\|_{\mathcal{G}} = v(r)^{-1}, \\ T(s, r) = T(r, s), \quad T(s, s) = 0, \end{cases} \tag{1}$$

where $\mathrm{grad}_s$ and $\mathrm{grad}_r$ denote the Riemannian gradients with respect to the source $s \in \mathcal{M}$ and the receiver $r \in \mathcal{M}$, respectively. The solution $T \colon \mathcal{M} \times \mathcal{M} \to \mathbb{R}_+$ corresponds to the travel-time function, and the interval $[v_{\min}, v_{\max}]$ specifies the minimum and maximum velocity values in the training set.

To prevent irregular behavior as $r \to s$, it is standard to factorize the travel-time function as

$$T(s,r) = \tilde{d}(s,r)\,\tau(s,r),$$

where $\tilde{d}$ is a *semimetric*—a function satisfying non-negativity ($\tilde{d}(s,r) \geq 0$), identity of indiscernibles ($\tilde{d}(s,r) = 0$ iff $s = r$), and symmetry ($\tilde{d}(s,r) = \tilde{d}(r,s)$)—that approximates the ground-truth Riemannian distance $d_{\mathcal{G}}$ [Smith et al., 2021, Waheed et al., 2021, Grubas et al., 2023, Li et al., 2024b, Kelshaw and Magri, 2024]. The scalar field $\tau(s,r)$ then represents the unknown travel-time factor.

In this work, we further require that $\tilde{d}(s,r)$ be invariant under the group action of $G$; that is, $\tilde{d}(s,r) = \tilde{d}(g \cdot s, g \cdot r)$ for all $g \in G$. This invariance condition is essential to ensure that the travel-time function preserves its steerability property, as will be demonstrated in Section 4. When the group acts by isometries, $\tilde{d}$ can, as previously discussed, be taken as the geodesic distance on the manifold. In particular, for manifolds embedded in Euclidean space where the group action extends to isometries of the ambient Euclidean space, the Euclidean (chordal) distance offers a computationally efficient approximation [Kelshaw and Magri, 2024]. For more general group actions, one may instead adopt the discrete semimetric $\tilde{d}(s,r) = \mathbb{1}_{s \neq r}$, which equals 1 when $s \neq r$ and 0 otherwise. To maintain gradient compatibility during training, this discrete indicator can be incorporated using a straight-through estimator [Bengio et al., 2013].

## 4 Method

We introduce Equivariant Neural Eikonal Solver (E-NES), which extends Equivariant Neural Fields to efficiently solve eikonal equations by leveraging geometric symmetries. Our approach incorporates steerability constraints that enable generalization across group transformations without explicit data augmentation. We present the theoretical framework (Section 4.1), detail our equivariant architecture (Section 4.2), introduce a technique for computing fundamental joint-invariants (Section 4.3), and describe our physics-informed training methodology (Section 4.4).

### 4.1 Theoretical Framework

We extend the Equivariant Neural Field architecture introduced in Wessels et al. [2024] to represent solutions of the eikonal equation. Let $(\mathcal{M}, \mathcal{G})$ denote the input Riemannian manifold on which the eikonal equations are defined, and let $G$ be a Lie group acting regularly on $\mathcal{M}$. We introduce a conditioning variable, represented as a geometric point cloud $z = \{(g_i, \mathbf{c}_i)\}_{i=1}^N$, which consists of $N$ so-called *pose-context* pairs. Here, each $g_i \in G$ is referred to as a *pose*, and each corresponding $\mathbf{c}_i \in \mathbb{R}^d$ is the associated *context* vector. We will denote the space of pose-context pairs as the product manifold $\mathcal{Z} = G \times \mathbb{R}^d$, so that $z$ is an element of the power set $\mathscr{P}(\mathcal{Z})$. This representation naturally supports a $G$-group action defined by $g \cdot z = \{(g \cdot g_i, \mathbf{c}_i)\}_{i=1}^N$.

In the setting of the factored eikonal equation, consider a solution $T_l$ satisfying Equation (1) for the velocity field $v_l$. We associate this solution with the conditioning variable $z_l$, such that our conditional neural field $T_\theta(s,r; z_l) = \tilde{d}(s,r)\,\tau_\theta(s,r; z_l)$ is trained to approximate $T_\theta(s,r; z_l) \approx T_l(s,r)$, for all $s, r \in \mathcal{M}$. Here, $\theta$ denotes the network weights.

The steerability constraint, i.e.,

$$T_\theta(s,r; g \cdot z) = T_\theta(g^{-1} \cdot s, \, g^{-1} \cdot r; \, z) \quad \text{for all } (s,r,z) \in \mathcal{M} \times \mathcal{M} \times \mathscr{P}(\mathcal{Z}), \qquad (2)$$

incorporates equivariance, enabling the network to generalize across all transformations $g \in G$ without requiring explicit data augmentation, thus significantly enhancing data efficiency. Consequently, solving the eikonal equation for one velocity field automatically extends to its entire family under group actions, as illustrated in Figure 1. This property is formally stated in the following proposition:

**Definition 4.1** ($g$-steered metric). *For all $g \in G$, define the g-steered metric $\mathcal{G}^g : T\mathcal{M} \times T\mathcal{M} \to \mathbb{R}$ as:*

$$\mathcal{G}_p^g\,(\dot{u}, \dot{v}) := \mathcal{G}_{gp}\left((\mathrm{d}L_{g^{-1}}(g \cdot p))^*[\dot{u}], (\mathrm{d}L_{g^{-1}}(g \cdot p))^*[\dot{v}]\right) \quad \text{for } p \in \mathcal{M}, \text{ and } \dot{u}, \dot{v} \in T_p\mathcal{M},$$

*where $L_{g^{-1}} : \mathcal{M} \to \mathcal{M}$ is the diffeomorphism defined by $L_{g^{-1}}(p) = g^{-1} \cdot p$.*

**Proposition 4.1** (Steered Eikonal Solution). *Let $T_\theta : \mathcal{M} \times \mathcal{M} \times \mathscr{P}(\mathcal{Z}) \to \mathbb{R}_+$ be a conditional neural field satisfying the steerability property* (2)*, and let $z_l$ be the conditioning variable representing the solution of the eikonal equation for $v_l : \mathcal{M} \to \mathbb{R}_+^*$, i.e., $T_\theta(s, r; z_l) \approx T_l(s, r)$ for $T_l$ satisfying Equation* (1) *for the velocity field $v_l$. Let $\mathcal{G}^g$ be a g-steered metric (Definition 4.1). Then:*

1. *The map $\mu : G \times (\mathcal{M} \to \mathbb{R}_+^*) \to (\mathcal{M} \to \mathbb{R}_+^*)$ defined by*

$$\mu(g, v_l)(s) := \left\| \mathrm{grad}_{g^{-1}s} \, T_l(g^{-1} \cdot s, \; g^{-1} \cdot r) \right\|_{\mathcal{G}^g}^{-1}, \tag{3}$$

*where $r$ is an arbitrary point in $\mathcal{M}$, **is a well-defined group action**.*

2. *For any $g \in G$, $T_\theta(s, r; g \cdot z_l)$ **solves the eikonal equation** with velocity field $\mu(g, v_l)$.*

For the common cases where the group action is either isometric or conformal, the expression for the associated velocity fields admits a simpler form:

**Corollary 4.1.** *Assume the hypotheses of Proposition 4.1, then the group action $\mu : G \times (\mathcal{M} \to \mathbb{R}_+^*) \to (\mathcal{M} \to \mathbb{R}_+^*)$ is given by:*

1. *$\mu(g, v_l)(s) = v_l(g^{-1} \cdot s)$ if $G$ acts isometrically on $\mathcal{M}$.*

2. *$\mu(g, v_l)(s) = \Omega(g, s) \, v_l(g^{-1} \cdot s)$ if $G$ acts conformally on $\mathcal{M}$ with conformal factor $\Omega(g, s) > 0$, i.e., $\mathcal{G}_{gs}(\mathrm{d}L_g(s)[\dot{s}_1], \mathrm{d}L_g(s)[\dot{s}_2]) = \Omega(g, s)^2 \, \mathcal{G}_s(\dot{s}_1, \dot{s}_2), \forall \dot{s}_1, \dot{s}_2 \in T_s\mathcal{M}$.*

Since $\mu : G \times (\mathcal{M} \to \mathbb{R}_+^*) \to (\mathcal{M} \to \mathbb{R}_+^*)$ constitutes a group action on the space of velocity fields, its orbit space induces a partition. Therefore, by obtaining the conditioning variable associated with one representative of an orbit, we effectively learn to solve the eikonal equation for all velocities within that equivalence class.

Finally, steerability also relates $\mathrm{grad}_s \, T_\theta(s, r; z)$ to $\mathrm{grad}_s \, T_\theta(s, r; g \cdot z)$ (see Lemma A.1). Hence, any geodesic extracted by backtracking the gradient of $T_\theta$ for one field generalizes to its transformed counterpart. This property is essential for applications such as geodesic segmentation [Chen and Cohen, 2019], motion planning [Ni and Qureshi, 2023], and ray tracing [Abgrall and Benamou, 1999].

Further details regarding the steerability property for eikonal equations, as well as proofs for Proposition 4.1 and Corollary 4.1, can be found in the Appendix (Section A).

## 4.2 Model Architecture

We define the Equivariant Neural Eikonal Solver (`E-NES`) as $\tau_\theta = P \circ E$, where .Wessels et al. [2024] and $P : \mathbb{R}^L \to \mathbb{R}_+$ is the bounded projection head from Grubas et al. [2023].

**1. Invariant Cross-Attention Encoder.** To enforce the steerability, our encoder builds invariant representations under $G$-symmetries of $(s, r, g_i)$. For each $g_i \in z$, we compute:

$$\mathbf{a}_i^{(s,r)} = \mathrm{RFF}\big(\mathrm{Inv}(s, r, g_i)\big), \quad \mathbf{a}_i^{(r,s)} = \mathrm{RFF}\big(\mathrm{Inv}(r, s, g_i)\big), \tag{4}$$

where $\mathrm{Inv}(\cdot)$ yields a complete set of functionally independent invariants via the moving frame method (as we will explain in Section 4.3), and $\mathrm{RFF}$ is a random Fourier feature mapping [Tancik et al., 2020]. To enforce $\tau_\theta(s, r; z) = \tau_\theta(r, s; z)$, we use $\tilde{\mathbf{a}}_i = (\mathbf{a}_i^{(s,r)} + \mathbf{a}_i^{(r,s)})/2$, the Reynolds operator over $S_2$ [Dym et al., 2024]. Then the invariant cross-attention encoder is computed as:

$$E(s, r; z) = \mathrm{FFN}_E \left( \sum_{i=1}^{N} \alpha_i \, v(\tilde{\mathbf{a}}_i, \mathbf{c}_i) \right) \quad \text{with} \quad \alpha_i = \frac{\exp(q(\tilde{\mathbf{a}}_i)^\top k(\mathbf{c}_i)/\sqrt{d_k})}{\sum_{j=1}^{N} \exp(q(\tilde{\mathbf{a}}_j)^\top k(\mathbf{c}_j)/\sqrt{d_k})},$$

where the attention maps and values are parameterized as:

$$q(\tilde{\mathbf{a}}) = W_q \tilde{\mathbf{a}}, \quad k(\mathbf{c}) = W_k \, \mathrm{LN}(W_c \mathbf{c}),$$
$$v(\tilde{\mathbf{a}}, \mathbf{c}) = \mathrm{FFN}_v(W_v \, \mathrm{LN}(W_c \mathbf{c}) \odot (1 + \mathrm{FFN}_\gamma(\tilde{\mathbf{a}})) + \mathrm{FFN}_\beta(\tilde{\mathbf{a}})),$$

with $\mathrm{FFN}_E, \mathrm{FFN}_v, \mathrm{FFN}_\gamma, \mathrm{FFN}_\beta$ being small multilayer perceptron (MLP) using GELU activation functions, and $W_q, W_k, W_c, W_v, W_\gamma, W_\beta$ learnable linear maps.

**2. Bounded Velocity Projection.** The encoder output $\mathbf{h} = E(s, r; z)$ passes through a second MLP network $\text{FFN}_P$ with AdaptiveGauss activations to model sharp wavefronts and caustics [Grubas et al., 2023]. The final output is projected into $[1/v_{\max}, 1/v_{\min}]$ by:

$$P(\mathbf{h}) = \left( \frac{1}{v_{\min}} - \frac{1}{v_{\max}} \right) \sigma(\alpha_0 \cdot \text{FFN}_P(\mathbf{h})) + \frac{1}{v_{\max}},$$

where $\sigma$ is the sigmoid function and $\alpha_0 \in \mathbb{R}_+$ is a learnable temperature parameter.

### 4.3 Computation of Fundamental Joint-Invariants

Let $\Pi = \mathcal{M}_1 \times \cdots \times \mathcal{M}_m$ denote a product of Riemannian manifolds, each equipped with a smooth, regular action $\delta_i : G \times \mathcal{M}_i \to \mathcal{M}_i$ by a Lie group $G$. These individual actions induce a natural diagonal action on the product $\Pi$ given by $\delta(g, (p_1, \ldots, p_m)) = (\delta_1(g, p_1), \ldots, \delta_m(g, p_m))$.

As observed in Olver [2001], when the group action on $\Pi$ is not free, the standard moving frame method is not directly applicable. In such cases, alternative techniques—such as those discussed in Section 2—are typically employed to compute invariants.

We show that the moving frame method can be restored in this setting by augmenting the space $\Pi$ with an auxiliary (learnable) group element, yielding an extended space $\overline{\Pi} = \Pi \times G$. On this augmented space, the group action admits a canonicalization procedure with explicitly computable invariants:

**Theorem 4.1** (Canonicalization via latent-pose extension). *Let $\Pi$ and $G$ be as above. Define a new group action $\bar{\delta} : G \times \overline{\Pi} \to \overline{\Pi}$ by $\bar{\delta}(h, (p_1, \ldots, p_m, g)) = (\delta(h, (p_1, \ldots, p_m)), h \cdot g)$. Then, the set $\left\{ \delta_i(g^{-1}, p_i) \right\}_{i=1}^m$ forms a complete collection of functionally independent invariants of the action $\overline{\mu}$.*

***Sketch of proof*** *(full at Appendix, Section B).* To verify that the action $\overline{\mu}$ is free, we show that the isotropy group of any point $(p_1, \ldots, p_m, g) \in \overline{\Pi}$ is trivial. Specifically, this subgroup satisfies $G_{(p_1, \ldots, p_m, g)} = G_{p_1} \cap \cdots \cap G_{p_m} \cap G_g$, where $G_{p_i}$ denotes the isotropy subgroup of $p_i$ under $\delta_i$, and $G_g$ is the isotropy subgroup of $g \in G$ under left multiplication. Since $h \cdot g = g$ implies $h = e$ in a group, we have $G_g = \{e\}$. Thus, the intersection is trivial, and $\bar{\delta}$ defines a free action. The moving frame method then guarantees a complete set of invariants, which are exactly $\{\delta_i(g^{-1}, p_i)\}_{i=1}^m$. $\quad \square$

This result formally justifies the construction proposed in Wessels et al. [2024], showing that the method yields a complete set of functionally independent invariants and thus guarantees full expressivity. Moreover, it extends the applicability of Equivariant Neural Fields to settings where $G$ acts regularly—but not necessarily freely nor transitive—on product manifolds. In particular, the invariant computation used in our E-NES architecture, as presented in Equation (4), takes the form

$$\text{Inv}(s, r, g_i) = ( g_i^{-1} \cdot s, g_i^{-1} \cdot r) \in \mathcal{M} \times \mathcal{M}.$$

### 4.4 Training details

Let $\mathcal{V} = \{v_l : \mathcal{M} \to [v_{\min}, v_{\max}]\}_{l=1}^K$ be our training set of $K$ velocity fields over the domain $\mathcal{M}$. At each iteration, we sample a batch $\mathcal{B}$ with $B$ velocity fields $\{v_i\}_{i=1}^B \subseteq \mathcal{V}$ and $N_{sr}$ source–receiver pairs $\{(s_{i,j}, r_{i,j})\}_{j=1}^{N_{sr}} \subset \mathcal{M}^2$ for each $v_i$. Let $\{z_i\}_{i=1}^B$ be the conditioning variables associated with $\{v_i\}_{i=1}^B$. To enforce the Eikonal equation, we express it in Hamiltonian form as $\mathcal{H}(s, r, T) = v(s)^2 \| \text{grad}_s T(s, r) \|_{\mathcal{G}}^2 - 1$, where the Eikonal equation is satisfied when $\mathcal{H} = 0$ [Grubas et al., 2023]. We then minimize a physics-informed loss that penalizes deviations from this zero-level set at both source and receiver locations:

$$L(\theta, \{z_i\}_{i=1}^B, \mathcal{B}) = \frac{1}{B N_{sr}} \sum_{i=1}^B \sum_{j=1}^{N_{sr}} \Big( \Big| v_i(s_{i,j})^2 \| \text{grad}_s T_\theta(s_{i,j}, r_{i,j}; z_i) \|_{\mathcal{G}}^2 - 1 \Big|$$
$$+ \Big| v_i(r_{i,j})^2 \| \text{grad}_r T_\theta(s_{i,j}, r_{i,j}; z_i) \|_{\mathcal{G}}^2 - 1 \Big| \Big). \tag{5}$$

Fitting is performed in the two modes presented in Wessels et al. [2024]. The first one is Autodecoding [Park et al., 2019] – where $z_l$ and $\theta$ are optimised simultaneously over a dataset. The second one is Meta-learning [Tancik et al., 2021, Cheng and Alkhalifah, 2024] – where optimization is split into an outer and inner loop, with $\theta$ being optimized in the outer loop and $z$ being re-initialized every outer step to solve the eikonal equation of the velocity fields in the batch in a limited number of SGD steps in the inner loop. We refer the readers to the Appendix (Section C) for further details.

Table 1: Performance comparison on OpenFWI datasets against FC-DeepONet. Colours denote **Best**, Second best, and Third best performing setups for each dataset. Fitting time represents the total computational time required to fit the latent conditioning variables for all 100 testing velocity fields.

| | FC-DeepONet | | E-NES Autodecoding (100 epochs) | | Autodecoding (convergence) | | Meta-learning | |
|---|---|---|---|---|---|---|---|---|
| Dataset | RE ($\downarrow$) | Fitting (s) | RE ($\downarrow$) | Fitting (s) | RE ($\downarrow$) | Fitting (s) | RE ($\downarrow$) | Fitting (s) |
| FlatVel-A | **0.00277** | $\sim 0.615$ | 0.00952 | 223.31 | 0.00506 | 1120.25 | 0.01065 | 5.92 |
| CurveVel-A | 0.01878 | $\sim 0.615$ | 0.01348 | 222.72 | **0.00955** | 1009.67 | 0.02196 | 5.91 |
| FlatFault-A | **0.00514** | $\sim 0.615$ | 0.00857 | 222.61 | 0.00568 | 1014.45 | 0.01372 | 5.92 |
| CurveFault-A | 0.00963 | $\sim 0.615$ | 0.01108 | 222.89 | **0.00820** | 1123.90 | 0.02086 | 5.92 |
| Style-A | 0,03461 | $\sim 0.615$ | 0.01034 | 222.00 | **0.00833** | 1117.99 | 0.01317 | 5.92 |
| FlatVel-B | **0.00711** | $\sim 0.615$ | 0.01581 | 222.74 | 0.00860 | 1010.32 | 0.02274 | 5.91 |
| CurveVel-B | 0.03410 | $\sim 0.615$ | 0.03203 | 222.97 | **0.02250** | 1127.87 | 0.03583 | 5.90 |
| FlatFault-B | 0.04459 | $\sim 0.615$ | 0.01989 | 222.70 | **0.01568** | 1133.98 | 0.03058 | 5.93 |
| CurveFault-B | 0.07863 | $\sim 0.615$ | 0.02183 | 222.89 | **0.01885** | 893.84 | 0.03812 | 5.89 |
| Style-B | 0.03463 | $\sim 0.615$ | 0.01171 | 221.90 | **0.01069** | 896.06 | 0.01541 | 5.90 |

## 5 Experiments

We evaluate Equivariant Neural Eikonal Solvers (E-NES) on the 2D OpenFWI benchmark [Deng et al., 2022] and extend our analysis to 3D settings to assess scalability and spherical geometry to show its generalization capabilities. Implementation details are provided in the Appendix (Section D). The code, including the experiments, is provided in the previously-mentioned public repository.

### 5.1 Benchmark on 2D-OpenFWI

Following Mei et al. [2024], we utilize ten velocity field categories from OpenFWI: FlatVel-A/B, CurveVel-A/B, FlatFault-A/B, CurveFault-A/B, and Style-A/B, each defined on a $70 \times 70$ grid. We train E-NES on 500 velocity fields per category and evaluate on 100 test fields, positioning four equidistant source points at the top boundary and computing travel times to all receiver coordinates. Additional evaluations using a denser $14 \times 14$ source grid are presented in the Appendix (Section E.2).

Performance is quantified using relative error (RE) and relative mean absolute error (RMAE):

$$RE := \frac{1}{N_s} \sum_{i=1}^{N_s} \sqrt{\frac{\sum_{j=1}^{M_p} |T_j^i - \hat{T}_j^i|}{\sum_{j=1}^{M_p} |T_j^i|^2}}, \quad RMAE := \frac{1}{N_s} \sum_{i=1}^{N_s} \frac{\sum_{j=1}^{M_p} |T_j^i - \hat{T}_j^i|}{\sum_{j=1}^{M_p} |T_j^i|},$$

where $N_s$ represents the total number of samples, $M_p$ denotes the total number of evaluated source-receiver pairs, $T_j^i$ indicates the $j$-th point of the $i$-th ground truth travel time, and $\hat{T}$ represents the model's predicted travel times. The ground truth values are generated using the second-order factored Fast Marching Method [Treister and Haber, 2016].

#### 5.1.1 Impact of Steerable Geometric Conditioning

To empirically validate the theoretical benefits of equivariance in our formulation, we conducted a controlled ablation study comparing E-NES with equivariance ($\mathcal{Z} = SE(2) \times \mathbb{R}^c$) against a variant without equivariance constraints ($\mathcal{Z} \cong \mathbb{R}^c$) on the Style-B dataset. Figure 2 illustrates consistent performance advantages with equivariance, demonstrated by lower values in both Eikonal loss and mean squared error (MSE) throughout the training process. This empirical validation substantiates our theoretical motivation for incorporating explicit equivariance constraints into the model architecture.

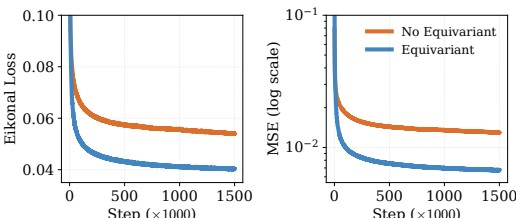

Figure 2: Comparative analysis of equivariant conditioning variables on the Style-B dataset. For non-equivariant models $\mathcal{Z} \cong \mathbb{R}^c$, while equivariant models use $\mathcal{Z} = SE(2) \times \mathbb{R}^c$.

To further assess the role of geometric conditioning, we compare our method against Functa [Dupont et al., 2022], a common baseline in the literature on conditional neural fields. Functa employs SIRENs [Sitzmann et al., 2020] with sample-specific scale and shift modulation but relies on global latent variables without geometric constraints, providing a clear contrast to our geometric point-cloud formulation of the conditioning variables. As shown in Table 2, our method consistently outperforms Functa across all datasets. We attribute this to Functa's global conditioning, which lacks localized representation and explicit geometric constraints.

### 5.1.2 Performance Comparison

Table 1 presents a systematic comparison between E-NES and FC-DeepONet [Mei et al., 2024] across all ten OpenFWI benchmark datasets. We evaluate three configurations of E-NES: autodecoding with 100 epochs (tradeoff between computational efficiency and performance), autodecoding until convergence (optimizing for accuracy), and meta-learning (prioritizing computational efficiency).

Our results demonstrate that E-NES with full autodecoding convergence outperforms FC-DeepONet in seven out of ten datasets, with particularly substantial improvements on the more challenging variants—Style-A/B, FlatFault-B, and CurveFault-B. Even with the reduced computational budget of 100 epochs, E-NES maintains competitive performance across most datasets. The meta-learning approach, while exhibiting moderately higher error rates, delivers remarkable computational efficiency—reducing fitting time from approximately 1000 seconds to under 6 seconds for the total 100 velocity fields, representing a two orders of magnitude improvement. Additional analyses are provided in Appendix F.3.

The quantitative results are supplemented by qualitative evaluations in the Appendix (Section E.6), including visualizations of travel-time predictions and spatial error distributions across all datasets. For a more detailed analysis of the trade-off between computational efficiency and prediction accuracy, including performance with varying numbers of autodecoding epochs, we refer to the ablation studies in the Appendix (Section E.5).

### 5.2 Extending to 3D: Scalability Analysis

Table 2: Performance comparison on OpenFWI B-type datasets against Functa. Fitting time represents the total computational time required to fit the latent conditioning variables for all 100 testing velocity fields. Here both methods perform 100 epochs of autodecoding to fit the latents.

| Dataset | Functa | | E-NES | |
|---|---|---|---|---|
| | RE ($\downarrow$) | Fitting (s) | RE ($\downarrow$) | Fitting (s) |
| FlatVel-B | 0.11854 | 12.55 | **0.01581** | 222.74 |
| CurveVel-B | 0.11210 | 12.49 | **0.03203** | 222.97 |
| FlatFault-B | 0.06428 | 12.50 | **0.01989** | 222.70 |
| CurveFault-B | 0.06146 | 12.72 | **0.02183** | 222.89 |
| Style-B | 0.03106 | 12.33 | **0.01171** | 221.90 |

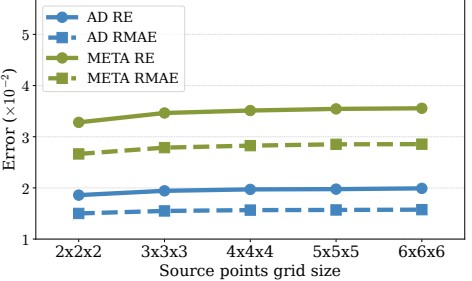

(a)

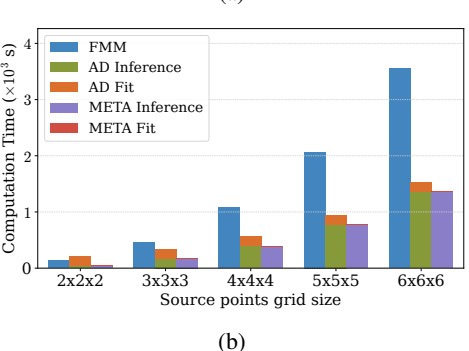

(b)

Figure 3: Scaling analysis of E-NES versus FMM on 3D OpenFWI data. (a) Both autodecoding and meta-learning maintain consistent error metrics (RE and RMAE, $\times 10^{-2}$) across increasing grid dimensions. (b) E-NES demonstrates computational advantages (seconds $\times 10^{3}$) over FMM even at minimal grid sizes, with efficiency gains amplifying as dimensions increase. Note that meta-learning fitting times (approximately 3 seconds) are barely visible in (b) due to their minimal magnitude relative to other displayed times.

To evaluate scalability to higher dimensions, we extended the Style-B dataset to 3D by extruding 2D velocity fields along the z-axis. Figure 3a shows that both autodecoding and meta-learning approaches maintain stable error metrics as grid dimensions increase, demonstrating E-NES's ability to model continuous fields independent of discretization resolution. Figure 3b shows E-NES maintains efficiency advantages over the Fast Marching Method (FMM) across all evaluated grid dimensions, with this advantage becoming more pronounced at larger scales (total computation time for all 100 test velocity fields). This stability stems from E-NES's continuous representation, which adapts to the underlying physics without requiring increasingly fine discretization.

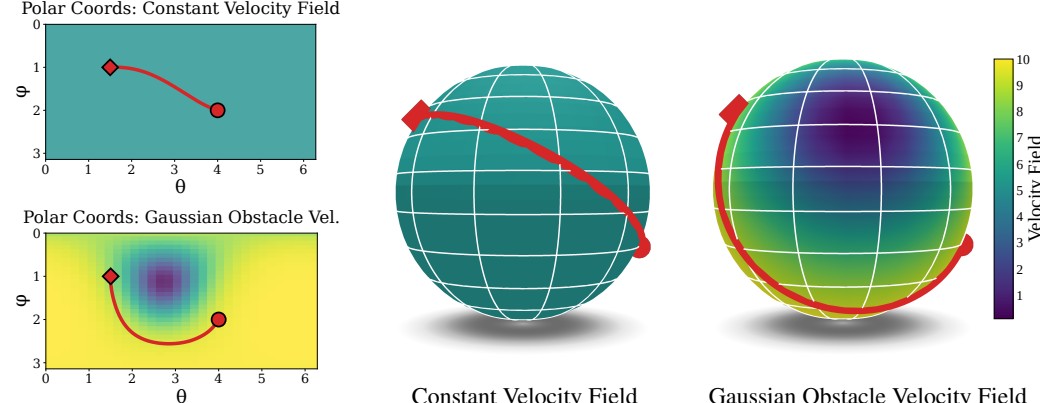

Figure 4: Geodesic path planning on the sphere using gradient integration of the travel-time function under two velocity fields. Left panels show the trajectories in local polar coordinates, while right panels visualize the corresponding paths on the spherical surface. The constant velocity field (top) yields a great-circle path, whereas the Gaussian obstacle velocity field (bottom) causes the trajectory to bend around the low-speed region. The diamond (◆) denotes the start and the circle (●) the goal.

## 5.3 Generalizability to Non-Euclidean Domains

We validate the generality of our framework on the 2-sphere, i.e., on $\mathbb{S}^2 \subset \mathbb{R}^3$, with $SO(2)$ steerability (rotations about the $z$-axis). This setting demonstrates two key capabilities: (i) handling non-transitive Lie group actions, and (ii) extending to non-Euclidean geometries.

Table 3: Performance of our method on Eikonal solvers over the 2-sphere.

| Dataset | RE (↓) | RMAE (↓) | Fitting Time (s) |
|---|---|---|---|
| Constant Speed | 0.013 | 0.012 | 209.2 |
| Spherical Style-B | 0.015 | 0.012 | 207.1 |

We test on two velocity field types: *constant speed* fields with velocities uniformly sampled, and *Spherical Style-B* fields, obtained by projecting OpenFWI's 2D Style-B fields onto the sphere via spherical coordinates. As shown in Table 3, our method achieves strong performance on both benchmarks, effectively learning the sphere's intrinsic geometry and correctly modeling wavefront propagation despite using Euclidean chordal distance $\tilde{d}(s, r)$ in the factorized representation (as described in Section 3.2).

Moreover, Figure 4 demonstrates that E-NES enables geodesic path planning via gradient integration, yielding optimal trajectories under configurations with and without obstacles. Additional details on how to perform this path-finding task are provided in the Appendix C.2.

## 6 Discussion and Future Work

In this work, we proposed a systematic approach to incorporate equivariance into neural fields and demonstrated its effectiveness through our Equivariant Neural Eikonal Solver (E-NES). Our experiments show that E-NES outperforms both Neural Operator methods (e.g., FC-DeepONet) and Conditional Neural Field approaches (e.g., Functa) across most benchmark datasets. The grid-free formulation is particularly advantageous for gradient integration tasks and naturally extends to Riemannian manifolds.

While our method requires explicit optimization at test time, FC-DeepONet's encoder forward pass performs implicit latent fitting (0.615 seconds for 100 velocity fields, as indicated in Table 1). Critically, our test-time optimization enables practitioners to dynamically adjust the accuracy-efficiency trade-off by varying the number of iterations (Appendix E.5). This adaptability parallels recent test-time optimization advances in large language models [Zhang et al., 2025], whereas FC-DeepONet's performance is fixed post-training. Additional comparative analyses are provided in Appendix F.1.

For future work, we plan to extend our analysis to homogeneous spaces beyond Euclidean and spherical domains, including position-orientation spaces for systems with nonholonomic constraints (e.g., vehicle path planning) and hyperbolic spaces for hierarchical interpolation tasks.

## Acknowledgements

We wish to thank Maksim Zhdanov for his help on the JAX implementation. Alejandro García Castellanos is funded by the Hybrid Intelligence Center, a 10-year programme funded through the research programme Gravitation which is (partly) financed by the Dutch Research Council (NWO). This publication is part of the project SIGN with file number VI.Vidi.233.220 of the research programme Vidi which is (partly) financed by the Dutch Research Council (NWO) under the grant `https://doi.org/10.61686/PKQGZ71565`. David Wessels is partially funded Ellogon.AI and a public grant of the Dutch Cancer Society (KWF) under subsidy (15059/2022-PPS2). Remco Duits and Nicky van den Berg gratefully acknowledge NWO for financial support via VIC- C 202.031.

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

# Appendix

## Table of Contents

# A  Steerability and Gradient Equivariance

Let $G$ be a Lie group acting smoothly on the Riemannian manifold $(\mathcal{M}, \mathcal{G})$ by left-translations

$$L_g \colon \mathcal{M} \to \mathcal{M}, \qquad L_g(p) = g \cdot p$$

and write $\mathrm{d}L_g(s) \colon T_s\mathcal{M} \to T_{gs}\mathcal{M}$ for its differential. Let $(\mathrm{d}L_g(s))^* \colon T_{gs}\mathcal{M} \to T_s\mathcal{M}$ denote the adjoint of $\mathrm{d}L_g(s)$ with respect to the metric $\mathcal{G}$.

**Lemma A.1** (Gradient Equivariance). *Let $T_\theta \colon \mathcal{M} \times \mathcal{M} \times \mathscr{P}(\mathcal{Z}) \to \mathbb{R}_+$ be a steerable conditional neural field, i.e. for all $g \in G$ and all $s, r \in \mathcal{M}$, $T_\theta(s, r; g \cdot z) = T_\theta(g^{-1} \cdot s, \ g^{-1} \cdot r; z)$. Then, for each fixed $z \in \mathcal{Z}$, fixed receiver $r \in \mathcal{M}$, and every $g \in G$,*

$$\mathrm{grad}_s \, T_\theta(s, r; g \cdot z) \ = \ (\mathrm{d}L_{g^{-1}}(s))^* \left[ \mathrm{grad}_{g^{-1}s} \, T_\theta(g^{-1} \cdot s, \ g^{-1} \cdot r; z) \right] \in T_s\mathcal{M}.$$

*Proof.* By steerability, one has

$$T_\theta(s, r; g \cdot z) = T_\theta\big(L_{g^{-1}}(s), \ L_{g^{-1}}(r); z\big).$$

Fix $r \in \mathcal{M}$ and differentiate with respect to $s$. For any $\dot{v} \in T_s\mathcal{M}$, the chain rule yields

$$\mathrm{d}_s T_\theta(s, r; g \cdot z)[\dot{v}] = \mathrm{d}_{g^{-1}s} T_\theta\big(g^{-1} \cdot s, \ g^{-1} \cdot r; z\big)\big[\mathrm{d}L_{g^{-1}}(s)[\dot{v}]\big].$$

By the defining property of the Riemannian gradient, we have $\mathcal{G}(\mathrm{grad}\, f, \cdot) = \mathrm{d}f$, so that:

$$\mathcal{G}_s\big( \underbrace{\mathrm{grad}_s \, T_\theta(s, r; g \cdot z)}_{\in T_s\mathcal{M}}, \dot{v}\big) = \mathcal{G}_{g^{-1}s}\big( \underbrace{\mathrm{grad}_{g^{-1}s}, T_\theta\big(g^{-1} \cdot s, g^{-1} \cdot r; z\big)}_{\in T_{g^{-1}s}\mathcal{M}}, \mathrm{d}L_{g^{-1}}(s)[\dot{v}]\big).$$

This exactly characterizes the adjoint $(\mathrm{d}L_{g^{-1}}(s))^*$, and the result follows. $\qquad\square$

**Definition 4.1 (Restated).** *For all $g \in G$, define the $g$-steered metric $\mathcal{G}^g \colon T\mathcal{M} \times T\mathcal{M} \to \mathbb{R}$ as:*

$$\mathcal{G}^g_p \, (\dot{u}, \dot{v}) := \mathcal{G}_{gp}\big((\mathrm{d}L_{g^{-1}}(g \cdot p))^*[\dot{u}], (\mathrm{d}L_{g^{-1}}(g \cdot p))^*[\dot{v}]\big) \quad \text{for } p \in \mathcal{M}, \text{ and } \dot{u}, \dot{v} \in T_p\mathcal{M}.$$

**Proposition 4.1 (Restated).** *Let $T_\theta \colon \mathcal{M} \times \mathcal{M} \times \mathscr{P}(\mathcal{Z}) \to \mathbb{R}_+$ be a conditional neural field satisfying the steerability property* (2), *and let $z_l$ be the conditioning variable representing the solution of the eikonal equation for $v_l \colon \mathcal{M} \to \mathbb{R}^*_+$, i.e., $T_\theta(s, r; z_l) \approx T_l(s, r)$ for $T_l$ satisfying Equation* (1) *for the velocity field $v_l$. Let $\mathcal{G}^g$ be a $g$-steered metric (Definition 4.1). Then:*

1. *The map $\mu \colon G \times (\mathcal{M} \to \mathbb{R}^*_+) \to (\mathcal{M} \to \mathbb{R}^*_+)$ defined by*

$$\mu(g, v_l)(s) := \frac{1}{\big\|\mathrm{grad}_{g^{-1}s} \, T_l(g^{-1} \cdot s, \ g^{-1} \cdot r)\big\|_{\mathcal{G}^g}}, \tag{6}$$

   *where $r$ is an arbitrary point in $\mathcal{M}$, **is a well-defined group action**.*

2. *For any $g \in G$, $T_\theta(s, r; g \cdot z_l)$ **solves the eikonal equation** with velocity field $\mu(g, v_l)$.*

*Proof.* By the steerability of $T_\theta$, for every $g \in G$ and $s, r \in \mathcal{M}$ we have

$$T_\theta(s, r; g \cdot z_l) = T_\theta(g^{-1} \cdot s, g^{-1} \cdot r; z_l).$$

Since $T_\theta(s, r; z_l) \approx T_l(s, r)$, it follows that

$$T_\theta(s, r; g \cdot z_l) \approx T_l(g^{-1} \cdot s, g^{-1} \cdot r).$$

Define the steered arrival time

$$T_l^g(s, r) := T_l(g^{-1} \cdot s, g^{-1} \cdot r).$$

We aim to show that $T_l^g$ satisfies the eikonal equation with velocity field $\mu(g, v_l)$.

**Gradient transformation.** By Lemma A.1, the gradient of $T_l^g$ is related to that of $T_l$ via

$$\operatorname{grad}_s T_l^g(s,r) = (\mathrm{d}L_{g^{-1}}(s))^* \left[ \operatorname{grad}_{g^{-1}s} T_l(g^{-1} \cdot s,\ g^{-1} \cdot r) \right].$$

Fix $g \in G$ and write $\dot{w} = \operatorname{grad}_{g^{-1}s} T_l(g^{-1} \cdot s, g^{-1} \cdot r) \in T_{g^{-1}s}\mathcal{M}$. Taking the squared $\mathcal{G}$-norm we get:

$$\| \operatorname{grad}_s T_l^g(s,r) \|_{\mathcal{G}}^2 = \mathcal{G}_s \left( (\mathrm{d}L_{g^{-1}}(s))^*[\dot{w}], (\mathrm{d}L_{g^{-1}}(s))^*[\dot{w}] \right).$$

Then, for $\mathcal{G}^g$ defined by Definition 4.1:

$$
\begin{aligned}
\| \operatorname{grad}_{g^{-1}s} T_l(g^{-1} \cdot s, g^{-1} \cdot r) \|_{\mathcal{G}^g}^2 &= \mathcal{G}_{g^{-1}s}^g (\dot{w}, \dot{w}) \\
&= \mathcal{G}_{gg^{-1}s} \left( (\mathrm{d}L_{g^{-1}}(gg^{-1} \cdot s))^*[\dot{w}], (\mathrm{d}L_{g^{-1}}(gg^{-1} \cdot s))^*[\dot{w}] \right) \\
&= \mathcal{G}_s \left( (\mathrm{d}L_{g^{-1}}(s))^*[\dot{w}], (\mathrm{d}L_{g^{-1}}(s))^*[\dot{w}] \right) \\
&= \| \operatorname{grad}_s T_l^g(s,r) \|_{\mathcal{G}}^2.
\end{aligned}
$$

By Eq. (3), we now get

$$\| \operatorname{grad}_s T_l^g(s,r) \|_{\mathcal{G}} = \frac{1}{\mu(g,v_l)(s)},$$

i.e., $T_l^g$ solves the eikonal equation with velocity $\mu(g, v_l)$.

**Group action properties of $\mu$.**
**(1) Identity:** Let $e \in G$ denote the identity. Then $e^{-1} = e$ and $\mathrm{d}L_e(s) = \mathrm{Id}$, hence:

$$\mathcal{G}_s^e = \mathcal{G}_s, \qquad \mu(e, v_l)(s) = \| \operatorname{grad}_s T_l(s,r) \|_{\mathcal{G}}^{-1} = v_l(s),$$

using the eikonal equation for $T_l$.

**(2) Compatibility:** For all $g, h \in G$, we show that:

$$\mu(g, \mu(h, v_l)) = \mu(gh, v_l).$$

We note that left and right hand side are respectively given by

$$\mu(g, \mu(h, v_l)) = \frac{1}{\| \operatorname{grad}_{h^{-1}g^{-1}s} T_l(h^{-1}g^{-1} \cdot s, h^{-1}g^{-1} \cdot r) \|_{(\mathcal{G}^h)^g}} \tag{7}$$

and

$$\mu(gh, v_l) = \frac{1}{\| \operatorname{grad}_{(gh)^{-1}s} T_l((gh)^{-1} \cdot s, (gh)^{-1} \cdot r) \|_{\mathcal{G}^{gh}}}. \tag{8}$$

Both equations (7) and (8) are the same iff $\mathcal{G}^{gh} = (\mathcal{G}^h)^g$. By Definition 4.1, it suffices to show $(\mathrm{d}L_{(gh)^{-1}})^* = (\mathrm{d}L_{g^{-1}})^*(\mathrm{d}L_{h^{-1}})^*$ which follows readily:

$$(\mathrm{d}L_{(gh)^{-1}})^* = (\mathrm{d}L_{h^{-1}} \circ \mathrm{d}L_{g^{-1}})^* = (\mathrm{d}L_{g^{-1}})^*(\mathrm{d}L_{h^{-1}})^*.$$

Hence, the group action $\mu$ is compatible.

$\square$

**Corollary 4.1 (Restated).** *Assume the hypotheses of Proposition 4.1, then the group action $\mu : G \times (\mathcal{M} \to \mathbb{R}_+^*) \to (\mathcal{M} \to \mathbb{R}_+^*)$ is given by:*

1. *$\mu(g, v_l)(s) = v_l(g^{-1} \cdot s)$ if $G$ acts isometrically on $\mathcal{M}$.*

2. *$\mu(g, v_l)(s) = \Omega(g, s)\, v_l(g^{-1} \cdot s)$ if $G$ acts conformally on $\mathcal{M}$ with conformal factor $\Omega(g, s) > 0$, i.e., $\mathcal{G}_{gs}(\mathrm{d}L_g(s)[\dot{s}_1], \mathrm{d}L_g(s)[\dot{s}_2]) = \Omega(g, s)^2\, \mathcal{G}_s(\dot{s}_1, \dot{s}_2), \forall \dot{s}_1, \dot{s}_2 \in T_s\mathcal{M}.$*

*Proof.* **(1) Isometric case:** If $G$ is an isometry, then $(\mathrm{d}L_{g^{-1}}(s))^* = (\mathrm{d}L_{g^{-1}}(s))^{-1} = \mathrm{d}L_g(g^{-1} \cdot s)$ – so that $\mathcal{G}^p$ is equal to the pull-back metric – and preserves inner products. Hence,

$$\| \dot{s} \|_{\mathcal{G}^g}^2 = \mathcal{G}_s^g(\dot{s}, \dot{s}) = \mathcal{G}_{gs}(\mathrm{d}L_g(g^{-1}g \cdot s)[\dot{s}], \mathrm{d}L_g(g^{-1}g \cdot s)[\dot{s}]) = \mathcal{G}_s(\dot{s}, \dot{s}) = \| \dot{s} \|_{\mathcal{G}}^2.$$

Therefore,

$$\left\| \operatorname{grad}_s T_\theta(s,r;g\cdot z_l) \right\|_{\mathcal{G}} = \left\| \operatorname{grad}_{g^{-1}s} T_\theta(g^{-1}\cdot s, g^{-1}\cdot r; z_l) \right\|_{\mathcal{G}^g} = \frac{1}{v_l(g^{-1}\cdot s)},$$

so $\mu(g,v_l)(s) = v_l(g^{-1}\cdot s)$.

**(2) Conformal case:** If $L_g$ acts conformally with factor $\Omega(g,s)$, then for all $\dot{s}_1, \dot{s}_2 \in T_s\mathcal{M}$,

$$\mathcal{G}_{gs}\left(\mathrm{d}L_g(s)[\dot{s}_1],\, \mathrm{d}L_g(s)[\dot{s}_2]\right) = \Omega(g,s)^2\, \mathcal{G}_s\left(\dot{s}_1,\, \dot{s}_2\right).$$

Hence $\mathrm{d}L_{g^{-1}}(s)$ scales lengths by $\Omega(g^{-1},s) = \Omega(g,s)^{-1}$ and its adjoint satisfies

$$\left(\mathrm{d}L_{g^{-1}}(s)\right)^* = \Omega(g,s)^{-2}\left(\mathrm{d}L_{g^{-1}}(s)\right)^{-1}, \quad \text{because } \left(\mathrm{d}L_g(s)\right)^* = \Omega(g,s)^2\left(\mathrm{d}L_g(s)\right)^{-1}.$$

Then

$$\operatorname{grad}_s T_\theta(s,r;g\cdot z_l) = \Omega(g,s)^{-2}\left(\mathrm{d}L_{g^{-1}}(s)\right)^{-1}[\operatorname{grad}_{g^{-1}s} T_\theta(g^{-1}\cdot s,\ g^{-1}\cdot r; z_l)],$$

and thus

$$\left\| \operatorname{grad}_s T_\theta(s,r;g\cdot z_l) \right\|_{\mathcal{G}} = \frac{1}{\Omega(g,s)\, v_l(g^{-1}\cdot s)}.$$

Therefore $\mu(g,v_l)(s) = \Omega(g,s)\, v_l(g^{-1}\cdot s)$. $\qquad\square$

As an example of an isometric action, consider 2D rotations with $G = SO(2)$ acting on $\mathcal{M} = \mathbb{R}^2$. In Figure 1, given a velocity field $v(s)$ and its corresponding conditioning variable $z$, the transformed variable $R_{\pi/6} \cdot z$ encodes the velocity field $v(R_{5\pi/6}s)$. Similarly, as an example of a conformal action, consider the positive scaling group $G = \mathbb{R}^*_+$ acting on $\mathcal{M} = \mathbb{R}$. In the graph shown in Figure 5, given a velocity field $v(s)$ encoded by $z$, the scaled conditioning variable $2\cdot z$ encodes the velocity field $2\,v(s/2)$. Additional examples on the steerability property as well as the emperical validation on our implementation can be found in Appendix E.7.

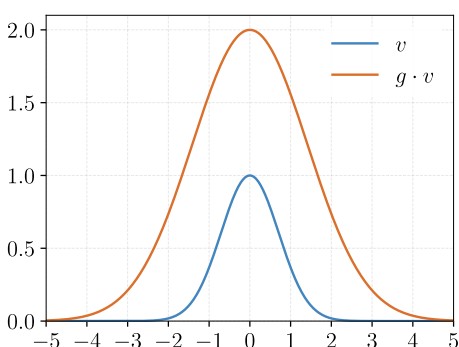

Figure 5: Example of conformal group action on $v(s) = e^{-s^2}$.

**Remark A.1** (Implementation in Euclidean Coordinates)**.** *In local coordinates, the metric tensor $\mathcal{G}_p$ at any point $p \in \mathcal{M}$ is represented by a symmetric positive-definite matrix, which induces an inner product on the tangent space $T_p\mathcal{M}$. Specifically, for any tangent vectors $\dot{p}_1, \dot{p}_2 \in T_p\mathcal{M}$, the inner product is given by $\mathcal{G}_p(\dot{p}_1,\dot{p}_2) = \langle \dot{p}_1, \dot{p}_2 \rangle_{\mathcal{G}} = \dot{p}_1^\top \mathcal{G}_p \dot{p}_2$, and the corresponding norm is defined as $\|\dot{p}_1\|_{\mathcal{G}} = \sqrt{\langle \dot{p}_1, \dot{p}_1 \rangle_{\mathcal{G}}}$.*

*Moreover, in local coordinates, the Riemannian gradient is related to that of the Euclidean gradient $\nabla f$ via the inverse metric tensor: $\operatorname{grad} f(p) = \mathcal{G}_p^{-1}\nabla f(p)$ [Absil et al., 2008]. If $\mathcal{M}$ is embedded in a Euclidean space, then the norm of the Riemannian gradient can be computed as $\|\operatorname{grad} f\|_{\mathcal{G}} = \|\nabla f\|_{\mathcal{G}^{-1}}$, and the adjoint can be computed as $(\mathrm{d}L_{g^{-1}}(s))^* = \mathcal{G}_s^{-1}(\mathrm{d}L_{g^{-1}}(s))^T \mathcal{G}_{g^{-1}s}$.*

*Therefore, the group action $\mu : G \times (\mathcal{M} \to \mathbb{R}^*_+) \to (\mathcal{M} \to \mathbb{R}^*_+)$ in local coordinates is given by:*

$$\mu(g,v_l)(s) = \left\| \nabla_{g^{-1}s} T_l\left(g^{-1}\cdot s, g^{-1}\cdot r\right) \right\|_{\widetilde{\mathcal{G}}^g}^{-1},$$

*where $r$ is an arbitrary point in $\mathcal{M}$, and $\widetilde{\mathcal{G}}^g : T_p\mathcal{M} \times T_p\mathcal{M} \to \mathbb{R}$ is the local coordinated expression of the g-steered metric given by*

$$\widetilde{\mathcal{G}}_p^g := \mathrm{d}L_{g^{-1}}(g\cdot p)\mathcal{G}_{gp}^{-1}(\mathrm{d}L_{g^{-1}}(g\cdot p))^T \quad \text{for any } p \in \mathcal{M}.$$

**Remark A.2** (Relation to pull-back metric)**.** *If $\mathcal{G}$ is isometric or conformal, then the g-steered metric is the pull-back metric $\mathcal{G}^g = (L_g)^*\mathcal{G}$ and one directly has*

$$\mathcal{G}^{g_1 g_2} = (\mathcal{G}^{g_2})^{g_1} \Rightarrow \mu(g_1 g_2, v_l) = \mu(g_1, \mu(g_2, v_1)).$$

# B Computing Invariants via Moving Frame

## B.1 Preliminaries

In this section, we present the essential mathematical foundations of the Moving Frame method, following Olver [2001], Sangalli et al. [2023]. For a comprehensive treatment, we refer readers to Olver [2011, 1995].

Let $\mathcal{M}$ be an $m$-dimensional smooth manifold and $G$ be an $r$-dimensional Lie group acting on $\mathcal{M}$. A (right) *moving frame* is a smooth equivariant map $\rho : \mathcal{M} \to G$ satisfying $\rho(g \cdot p) = \rho(p) \cdot g^{-1}$ for all $g \in G$ and $p \in \mathcal{M}$. Every moving frame $\rho$ induces a canonicalization function $k : \mathcal{M} \to \mathcal{M}$ defined by

$$k(p) = \rho(p) \cdot p,$$

which is $G$-invariant:

$$\forall p \in \mathcal{M}, \ g \in G, \quad k(g \cdot p) = \rho(g \cdot p) \cdot g \cdot p = \rho(p) \cdot p = k(p).$$

The existence of moving frames is characterized by the following fundamental result:

**Theorem B.1** (Existence of moving frame; see Olver [2001]). *A moving frame exists in a neighborhood of a point $p \in \mathcal{M}$ if and only if $G$ acts freely and regularly near $p$.*

Moving frames can be constructed via cross-sections to group orbits. A *cross-section* to the group orbits is a submanifold $K \subseteq \mathcal{M}$ of complementary dimension to the group (i.e., $\dim K = m - r$) that intersects each orbit transversally at exactly one point.

**Theorem B.2** (Moving frame from cross-section; see Olver [2001]). *If $G$ acts freely and regularly on $\mathcal{M}$, then given a cross-section $K$ to the group orbits, for each $p \in \mathcal{M}$ there exists a unique element $g_p \in G$ such that $g_p \cdot p \in K$. The function $\rho : \mathcal{M} \to G$ mapping $p$ to $g_p$ is a moving frame.*

While any regular Lie group action admits multiple local cross-sections, coordinate cross-sections (obtained by fixing $r$ of the coordinates) are particularly useful for determining fundamental invariants of $\mathcal{M}$ with regards to $G$.

**Theorem B.3** (Fundamental invariants via coordinate cross-sections; see Olver [2001]). *Given a free, regular Lie group action and a coordinate cross-section $K$, let $\rho$ be the associated moving frame. Then the non-constant coordinates of the canonicalization function image*

$$k(p) = \rho(p) \cdot p \in K$$

*form a complete system of functionally independent invariants.*

Moreover, this theorem aligns with the classical result on the structure and separation properties of invariants:

**Theorem B.4** (Existence of Fundamental Invariants; see [Olver, 1995]). *Let $G$ be a Lie group acting freely and regularly on an $m$-dimensional manifold $\mathcal{M}$ with orbits of dimension $s$. Then, for each point $p \in \mathcal{M}$, there exist $m - s$ functionally independent invariants $I_1, \ldots, I_{m-s}$ defined on a neighborhood $U$ of $p$ such that any other invariant $I$ on $U$ can be expressed as $I = H(I_1, \ldots, I_{m-s})$ for some function $H$. Moreover, these invariants separate orbits: two points $y, z \in U$ lie in the same orbit if and only if $I_v(y) = I_v(z)$ for all $v = 1, \ldots, m - s$.*

These foundational results—existence, construction via cross-sections, and the properties of invariants—underlie all moving-frame computations and guarantee that one obtains a complete, orbit-separating set of invariants.

## B.2 Canonicalization via latent-pose extension

**Theorem 4.1 (Restated).** *Let $G$ be a Lie group acting smoothly and regularly (but not necessarily freely) on each Riemannian manifold $\mathcal{M}_i$ via $\delta_i : G \times \mathcal{M}_i \to \mathcal{M}_i$, for $i = 1, \ldots, m$, and hence diagonally on*

$$\Pi = \mathcal{M}_1 \times \cdots \times \mathcal{M}_m, \quad \delta\big(g, (p_1, \ldots, p_m)\big) = \big(\delta_1(g, p_1), \ldots, \delta_m(g, p_m)\big).$$

*On the augmented space $\overline{\Pi} = \Pi \times G$, define $\bar{\delta}\big(h, (p_1, \ldots, p_m, g)\big) = \big(\delta(h, (p_1, \ldots, p_m)), \ hg\big).$*

*Then:*

1. *$\overline{\delta}$ is free.*

2. *A moving frame is given by $\rho : \overline{\overline{\Pi}} \to G$, such that $\rho(p_1, \ldots, p_m, g) = g^{-1}$.*

3. *The set $\left\{ \delta_i(g^{-1}, p_i) \right\}_{i=1}^m$ forms a complete collection of functionally independent invariants of the action $\overline{\mu}$.*

*Proof.*
**(1) Freeness:** Fix $(p_1, \ldots, p_m, g) \in \overline{\overline{\Pi}}$. Its isotropy subgroup is

$$G_{(p_1, \ldots, p_m, g)} = \left\{ h \in G : \ \delta(h, (p_1, \ldots, p_m)) = (p_1, \ldots, p_m), \ h\,g = g \right\}.$$

Hence

$$G_{(p_1, \ldots, p_m, g)} = G_{p_1} \cap \cdots \cap G_{p_m} \cap G_g.$$

where $G_{p_i}$ denotes the isotropy subgroup of $p_i$ under $\delta_i$, and $G_g$ is the isotropy subgroup of $g \in G$ under left multiplication. Since $hg = g$ implies $h = e$ in a group, we have $G_g = \{e\}$. Thus, the intersection is trivial, and $\overline{\mu}$ defines a free action.

**(2) Moving frame existence:** Because $\overline{\delta}$ is free and regular, Theorem B.1 guarantees a unique smooth equivariant map

$$\rho : \overline{\overline{\Pi}} \to G$$

associated to the cross-section $K = \{(p_1, \ldots, p_m, g) \in \overline{\overline{\Pi}} : g = e\}$. A direct check shows

$$\rho(p_1, \ldots, p_m, g) = g^{-1},$$

and one verifies equivariance

$$\rho\big(\overline{\delta}(h, (p_1, \ldots, p_m, g))\big) = (h\,g)^{-1} = g^{-1}h^{-1} = \rho(p_1, \ldots, p_m, g)\,h^{-1}.$$

**(3) Functional independence and completeness:** The canonicalization map $k(p_1, \ldots, p_m, g) = \overline{\delta}(\rho(p_1, \ldots, p_m, g), (p_1, \ldots, p_m, g)) \in K$ gives

$$\begin{aligned}
k(p_1, \ldots, p_m, g) &= \overline{\delta}(g^{-1}, (p_1, \ldots, p_m, g)) \\
&= \big(\delta_1(g^{-1}, p_1), \ldots, \delta_m(g^{-1}, p_m), \ e\big).
\end{aligned}$$

By Theorem B.3, the nonconstant coordinates $\delta_i(g^{-1}, p_i)$ are functionally independent and generate all invariants of $\overline{\delta}$. $\qquad\square$

## C  Training and Inference Details

### C.1  Training

**Autodecoding.** In autodecoding, we jointly optimize both the network parameters $\theta$ and the latent conditioning variables $z_i$ across the dataset. As detailed in Algorithm 1, this approach yields a tighter fit to the eikonal equation, at the expense of longer training times. Empirically, we observe that convergence on the validation set requires between 250 and 500 fitting epochs.

**Meta-learning.** Our meta-learning framework separates training into two loops: an inner loop for optimizing latents and an outer loop for updating network parameters. Algorithm 2 summarizes this bi-level optimization procedure.

By leveraging meta-learning, we achieve significantly faster fitting and impose additional structure on the latent space [Knigge et al., 2024]. However, as noted by Dupont et al. [2022], the small number of inner-loop updates typically used on meta-learning can restrict expressivity. To mitigate this, we initialize the network with weights pretrained via autodecoding, which accelerates convergence and often leads to better local minima (see Section E.1).

---

**Algorithm 1** Autodecoding Training

---

**Require:** Velocity fields $\mathcal{V} = \{v_l\}_{l=1}^{K}$, epochs $num\_epochs$, batch size $B$, pairs per field $N_{sr}$, learning rate $\eta$
 1: Randomly initialize shared base network $T_\theta$
 2: Initialize latents $z_l \leftarrow \{(g_i, \mathbf{c}_i)\}_{i=1}^{N}$ **for all velocity fields**
 3: **for** $epochs = 1$ to $num\_epochs$ **do**
 4:     **while** dataloader not empty **do**
 5:         Sample batch $\mathcal{B} = \{(s_{i,j}, r_{i,j}, v_i(s_{i,j}), v_i(r_{i,j}))\}_{i=1,j=1}^{B,N_{sr}}$
 6:         Compute loss $L(\theta, \{z_i\}_{i=1}^{B}, \mathcal{B})$ (see Equation 5)
 7:         Update $\theta \leftarrow \theta - \eta \nabla_\theta L$
 8:         Update each $z_i \leftarrow z_i - \eta \nabla_{z_i} L$
 9:     **end while**
10: **end for**
**Ensure:** Trained $\theta$ and latents $\{z_l\}_{l=1}^{K}$

---

---

**Algorithm 2** Meta-learning Training

---

**Require:** Velocity fields $\mathcal{V} = \{v_l\}_{l=1}^{K}$, outer epochs $num\_epochs$, inner steps $S$, batch size $B$, pairs per field $N_{sr}$, learning rates $\eta_\theta, \eta_{\text{SGD}}$
 1: Initialize shared base network $T_\theta$ (optionally pretrained), and learnable learning rate $\eta_z$.
 2: **for** $epochs = 1$ to $num\_epochs$ **do**
 3:     **while** dataloader not empty **do**
 4:         Sample batch of velocity fields $\{v_i\}_{i=1}^{B} \subseteq \mathcal{V}$
 5:         Initialize latents $z_i^{(0)}$ for each $v_i$
 6:         **for** $t = 1$ to $S$ **do**                        ▷ **Inner loop:** Update latents
 7:             Sample $N_{sr}$ source–receiver pairs $\{(s_{i,j}^{(t-1)}, r_{i,j}^{(t-1)})\}_{j=1}^{N_{sr}} \subset \mathcal{M}^2$, for each $v_i$
 8:             Construct batch $\mathcal{B}^{(t-1)} = \{(s_{i,j}^{(t-1)}, r_{i,j}^{(t-1)}, v_i(s_{i,j}^{(t-1)}), v_i(r_{i,j}^{(t-1)}))\}_{i=1,j=1}^{B,N_{sr}}$
 9:             Compute $\widetilde{L}(\theta, \{z_i^{(t-1)}\}_{i=1}^{B}, \mathcal{B}^{(t-1)})$
10:             Update each $z_i^{(t)} \leftarrow z_i^{(t-1)} - \eta_z \nabla_{z_i} \widetilde{L}$
11:         **end for**
12:         Sample $N_{sr}$ source–receiver pairs $\{(s_{i,j}^{(S)}, r_{i,j}^{(S)})\}_{j=1}^{N_{sr}} \subset \mathcal{M}^2$, for each $v_i$
13:         Construct batch $\mathcal{B}^{(S)} = \{(s_{i,j}^{(S)}, r_{i,j}^{(S)}, v_i(s_{i,j}^{(S)}), v_i(r_{i,j}^{(S)}))\}_{i=1,j=1}^{B,N_{sr}}$
14:         Compute $\widetilde{L}_{meta}(\theta) = \widetilde{L}(\theta, \{z_i^{(S)}\}_{i=1}^{B}, \mathcal{B}^{(S)})$
15:         Update $\theta \leftarrow \theta - \eta_\theta \nabla_\theta \widetilde{L}_{meta}$
16:         Update $\eta_z \leftarrow \eta_z - \eta_{\text{SGD}} \nabla_{\eta_z} \widetilde{L}_{meta}$
17:     **end while**
18: **end for**
**Ensure:** Trained $\theta$

---

In our implementation, we utilize an alternative loss function $\widetilde{L}$ in place of the original loss defined in Equation 5. Specifically, $\widetilde{L}$ employs the log-hyperbolic cosine loss $\log(\cosh(x))$ [Jeendgar et al., 2022], as a differentiable substitute for the absolute value term in Equation 5. This substitution is critical for effective meta-learning, as the log-cosh function provides a smooth approximation to the absolute value while maintaining convexity and ensuring well-behaved gradients throughout its domain. The differentiability properties of this function enable us to obtain high-quality higher-order derivatives, which are essential for the backpropagation process through all inner optimization steps, as outlined in lines 15 and 16 of Algorithm 2.

### C.2 Inference

**Solving for new velocity fields.** Given a new set of velocity fields, we will obtain their corresponding latent representation as follows:

- **Autodecoding:** we perform Algorithm 1 using the frozen weights $\theta^*$, i.e., we do not perform steps 1 and 7.

- **Meta-learning:** we perform the inner loop of the Algorithm 2 using frozen weights $\theta^*$, i.e., we do steps 5 to 11.

**Execution of bidirectional backward integration.** As stated in Section 4, given the solution of the eikonal equation, you can obtain the shortest path between two points under the given velocity via backward integration. Indeed, we perform SGD over the normalized gradients $\| \operatorname{grad}_s T(s,r)\|_{\mathcal{G}} \operatorname{grad}_s T(s,r)$ [Bekkers et al., 2015]. Furthermore, as shown in Ni and Qureshi [2023], our model's symmetry behavior allows us to perform gradient steps bidirectionally from source to receiver and from receiver to source. Hence, we compute the final path solution bidirectionally using iterative Riemannian Gradient Descent [Absil et al., 2008] by updating the source and receiver points as follows, where $\alpha \in \mathbb{R}$ is a step size hyperparameter.

$$\begin{cases} s^{(t)} \leftarrow R_{s^{(t-1)}}\left(-\alpha \| \operatorname{grad}_s T_{\theta^*}(s^{(t-1)}, r^{(t-1)}; z_l)\|_{\mathcal{G}} \operatorname{grad}_s T_{\theta^*}(s^{(t-1)}, r^{(t-1)}; z_l)\right), \\ r^{(t)} \leftarrow R_{r^{(t-1)}}\left(-\alpha \| \operatorname{grad}_r T_{\theta^*}(s^{(t-1)}, r^{(t-1)}; z_l)\|_{\mathcal{G}} \operatorname{grad}_r T_{\theta^*}(s^{(t-1)}, r^{(t-1)}; z_l)\right), \end{cases} \quad (9)$$

where $R_p : T_p\mathcal{M} \to \mathcal{M}$ is a *retraction* at $p \in \mathcal{M}$. The retraction mapping will provide a notion of moving in the direction of a tangent vector, while staying on the manifold:

**Definition C.1** (Retraction; see Absil et al. [2008]). *A retraction on a manifold $\mathcal{M}$ is a smooth mapping $R$ from the tangent bundle $T\mathcal{M}$ onto $\mathcal{M}$ with the following properties. Let $R_p$ denote the restriction of $R$ to $T_p\mathcal{M}$.*

*(i) $R_p(\dot{0}_p) = p$, where $\dot{0}_p$ denotes the zero element of $T_p\mathcal{M}$.*

*(ii) With the canonical identification $T_{\dot{0}_p} T_p\mathcal{M} \simeq T_p\mathcal{M}$, $R_p$ satisfies*

$$\mathrm{d}R_p(\dot{0}_p) = \mathrm{id}_{T_p\mathcal{M}},$$

*where $\mathrm{id}_{T_p\mathcal{M}}$ denotes the identity mapping on $T_p\mathcal{M}$.*

# D   Experimental Details

This section presents the comprehensive training and validation hyperparameters employed in the experiments described in Section 5. All experiments were conducted using a single NVIDIA H100 GPU.

## D.1   2D OpenFWI Experiments

**Model Architecture.** Our invariant cross-attention implementation utilizes a hidden dimension of 128 with 2 attention heads. The conditioning variables are defined as $z \in \mathcal{P}(\mathcal{Z})$ with cardinality $|z| = 9$, where $\mathcal{Z} = SE(2) \times \mathbb{R}^{32}$ for each velocity field. We initialize the pose component of the latents—derived from $SE(2) = \mathbb{R}^2 \times S^1$—at equidistant positions in $\mathbb{R}^2$, with orientations randomly sampled from a uniform distribution over $[-\pi, \pi)$. The context component of the latents is initialized as constant unit vectors.

For embedding the invariants, we employ RFF-Net, a variant of Random Fourier Features with trainable frequency parameters. This approach enhances training robustness with respect to hyperparameter selection. Following the methodology of Wessels et al. [2024], we implement two distinct RFF embeddings: one for the value function and another for the query function of the cross-attention mechanism. The frequency parameters are initialized to 0.05 for the query function and 0.2 for the value function.

Detailed ablations on these architectural choices can be found in Section E.3 and Section E.4

**Dataset Configuration.** For each OpenFWI dataset, we sample 600 velocity fields for training and 100 for validation. We further divide the training set into 500 fields for training and 100 fields for testing. For each velocity field, we uniformly sample 20,480 coordinates, producing 10,240 pairs per velocity field. Each batch comprises two velocity fields with 5,120 source-receiver pairs per field.

**Training Protocol.** The autodecoding phase consists of 3,000 epochs, while the meta-learning phase comprises 500 epochs. To mitigate overfitting, we report results based on the model that performs optimally on the validation set. Under this criterion, the effective training duration for autodecoding averages approximately 920 epochs (3.6 hours per dataset), while meta-learning averages 440 epochs (17.8 hours per dataset).

**SE(2) Optimization with Pseudo-Exponential Map.** For optimizing parameters in $SE(2) = \mathbb{R}^2 \times S^1$, we employ a standard simplification of Riemannian optimization for affine groups, known as parameterization via the "pseudo-exponential map." This approach substitutes the exponential map of $SE(n) = \mathbb{R}^n \rtimes SO(n)$ with the exponential map of $\mathbb{R}^n \times SO(n)$ [Solà et al., 2021, Claraco, 2022]. This technique is applied in both autodecoding and meta-learning phases, though with different optimizers as detailed below.

**Autodecoding Optimization Strategy.** For autodecoding, we optimize all parameters using the Adam optimizer with different learning rates for each component. The model parameters are trained with a learning rate of $10^{-4}$. For the latent variables, context vectors use a learning rate of $10^{-2}$, while pose components in $SE(2)$ are optimized with a learning rate of $10^{-3}$. Both latent variable components (context and pose) are optimized using Adam.

**Meta-learning Configuration.** For meta-learning, we jointly optimize the model parameters $\theta$ and inner loop learning rates $\eta_z$ using Adam with a cosine scheduler. This scheduler implements a single cycle with an initial learning rate of $10^{-4}$ and a minimum learning rate of $10^{-6}$. For the SGD inner loop optimization, we initialize the learning rates at 30 for context vectors and 2 for pose components, executing 5 optimization steps in the inner loop.

**Functa Model Architecture.** For the experiment presented in Table 2, adapt the model presented in Dupont et al. [2022] to have characteristics similar to our E–NES model. Specifically, we use a global conditioning variable of $z \in \mathbb{R}^{315}$ to match the total number of parameters in our geometric point cloud. Moreover, we will use a hidden dimension of 128 and a latent modulation size of 128 to match the dimensionality of our architecture.

## D.2 3D OpenFWI Experiments

**Model Architecture.** For the 3D experiments, we adapt the architecture described in the 2D case with several key modifications. Most notably, we reduce the cross-attention mechanism to a single head rather than the two employed in the 2D experiments. Additionally, we utilize a set of eight elements in $\mathcal{Z} = SE(3) \times \mathbb{R}^{32}$ as conditioning variables instead of the nine used in the 2D case.

**Pose Representation.** While we maintain the same general approach for pose optimization as in the 2D experiments, the 3D case requires parameterization of $SE(3)$ rather than $SE(2)$. We employ the pseudo-exponential map as described previously, but with an important distinction in the rotation component. Specifically, we parameterize the $SO(3)$ component using Euler angles.

**Training and Optimization.** All other aspects of the training procedure—including dataset configuration, optimization strategies, learning rates, and epoch counts—remain consistent with those detailed in the 2D experiments. We maintain the same distinction between optimization approaches in autodecoding (Adam for all parameters) and meta-learning (Adam for model parameters in the outer loop, SGD for latent variables in the inner loop). This consistency allows for direct comparison between 2D and 3D experimental results while accounting for the specific requirements of 3D modeling.

## D.3 Spherical Experiments

**Model Architecture.** For the spherical experiments, we adapt the architecture from the 2D Euclidean case with several key modifications. First, we reduce the cross-attention mechanism to a single head for the constant-velocity dataset, while using two heads for the non-constant cases as in the 2D experiments. Additionally, we employ conditioning variables from $\mathcal{Z} = SO(2) \times \mathbb{R}^{32}$ with nine elements for the non-constant case, compared to four elements from $SO(2) \times \mathbb{R}^{16}$ for the constant

case. These differences reflect the reduced representational requirements of the constant-velocity scenario, which exhibits simpler dynamics than its non-constant counterparts.

**Dataset Configuration.** We evaluate our method on three types of velocity fields defined over the sphere:

- **Constant Speed Fields:** For each sample, we draw a scalar velocity $v \sim \mathcal{U}(0.1, 2.0)$ and define a constant velocity field over the entire sphere.
- **Spherical Style-B Fields:** We construct spatially-varying velocity fields by projecting OpenFWI's 2D Style-B velocity models onto the sphere. We query the velocity field at continuous coordinates via RBF interpolation using cosine distance kernels.
- **Gaussian Obstacle Fields:** For each sample, we generate random von Mises-Fisher distributions on the sphere with concentration parameters sampled uniformly from $\kappa \sim \mathcal{U}(1, 5)$. The distributions are normalized to the interval $[0.1, 10.0]$ to enforce distinctions between low-velocity regions (obstacles) and high-velocity regions.

Ground truth travel times are computed using the Hamiltonian Fast Marching method of Mirebeau and Portegies [2019] with the canonical spherical metric tensor.

## E   Additional results

### E.1   Impact of Autodecoding Pretraining on Meta-Learning Performance

This section examines the effectiveness of initializing meta-learning with parameters derived from standard autodecoding pretraining. We present a comparative analysis using the Style-A and CurveVel-A 2D OpenFWI datasets, evaluating performance through both eikonal loss and mean squared error (MSE) metrics throughout the training process.

Figure 6 illustrates the training dynamics across both initialization strategies. Our results demonstrate that utilizing pretrained model parameters from the autodecoding phase substantially enhances convergence characteristics in two critical dimensions. First, pretrained initialization enables significantly faster convergence, reducing the number of required epochs to reach performance plateaus. Second, and more importantly, this approach allows the optimization process to achieve superior local minima compared to random initialization.

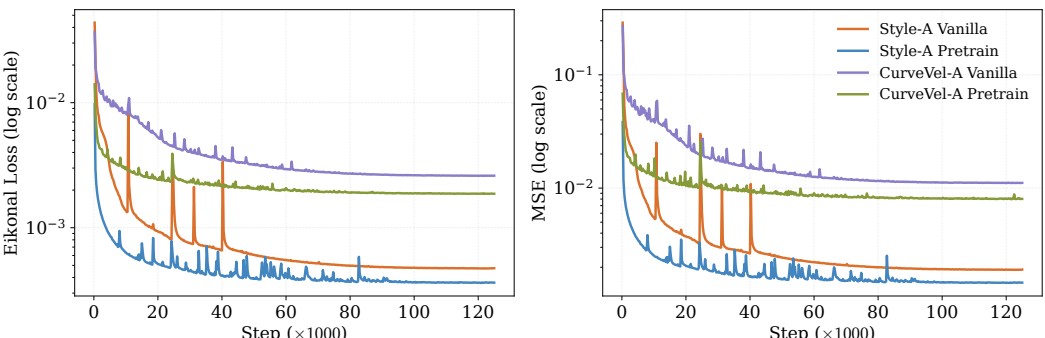

Figure 6: Comparative analysis of meta-learning convergence with pretrained versus random initialization on Style-A and CurveVel-A OpenFWI datasets.

These findings highlight a fundamental efficiency in our methodology: by leveraging pretrained autodecoding parameters, the meta-learning phase is effectively transformed from learning from scratch to a targeted adaptation task. Specifically, the pretrained model has already established a robust conditional neural field representation of the underlying physics. The subsequent meta-learning process then primarily needs to adapt this existing representation to interpret conditioning variables obtained through just 5 steps of SGD, rather than those refined over 500 epochs of autodecoding. This represents a significant computational advantage, as the meta-learning algorithm can focus exclusively on learning the mapping between rapidly-obtained SGD variables and the already-established neural field, rather than simultaneously learning both the field representation and the optimal conditioning.

## E.2 Comprehensive Grid Evaluation of OpenFWI Performance

In Section 5, we evaluated `E-NES` performance on the OpenFWI benchmark following the methodology established by Mei et al. [2024], which utilizes four equidistant source points at the top of the velocity fields and measures predicted travel times from these sources to all 70×70 receiver coordinates. This approach allows direct comparison with both the Fast Marching Method (FMM) and the FC-DeepOnet model [Mei et al., 2024].

To validate the robustness and generalizability of our approach, we conducted additional experiments using a substantially denser sampling protocol—specifically, a uniform 14×14 source point grid similar to that employed by Grubas et al. [2023]. As demonstrated in Table 4, `E-NES` maintains performance metrics comparable to those presented in Table 1, despite the significant increase in source points and resulting source-receiver pairs.

Table 4: Performance on OpenFWI datasets on a $14 \times 14$ grid of source points. Fitting time represents the total computational time required to fit the latent conditioning variables for all 100 testing velocity fields.

| | Autodecoding (100 epochs) | | | Autodecoding (convergence) | | | Meta-learning | | |
|---|---|---|---|---|---|---|---|---|---|
| Dataset | RE ($\downarrow$) | RMAE ($\downarrow$) | Fitting (s) | RE ($\downarrow$) | RMAE ($\downarrow$) | Fitting (s) | RE ($\downarrow$) | RMAE ($\downarrow$) | Fitting (s) |
| FlatVel-A | 0.01023 | 0.00827 | 223.31 | 0.00624 | 0.00509 | 1010.90 | 0.01304 | 0.01003 | 5.92 |
| CurveVel-A | 0.01438 | 0.01139 | 222.72 | 0.01069 | 0.00841 | 1009.67 | 0.02460 | 0.01878 | 5.91 |
| FlatFault-A | 0.01050 | 0.00751 | 222.61 | 0.00744 | 0.00510 | 1014.45 | 0.01749 | 0.01255 | 5.92 |
| CurveFault-A | 0.01380 | 0.00976 | 222.89 | 0.01088 | 0.00745 | 1007.97 | 0.02471 | 0.01807 | 5.92 |
| Style-A | 0.00962 | 0.00785 | 222.00 | 0.00795 | 0.00646 | 783.13 | 0.01326 | 0.01036 | 5.92 |
| FlatVel-B | 0.01988 | 0.01586 | 222.74 | 0.01178 | 0.00906 | 786.48 | 0.03077 | 0.02474 | 5.91 |
| CurveVel-B | 0.04291 | 0.03349 | 222.97 | 0.03297 | 0.02528 | 1010.70 | 0.04977 | 0.03930 | 5.90 |
| FlatFault-B | 0.01889 | 0.01413 | 222.70 | 0.01557 | 0.01147 | 898.28 | 0.02998 | 0.02214 | 5.93 |
| CurveFault-B | 0.02244 | 0.01728 | 222.89 | 0.01991 | 0.01537 | 561.22 | 0.03824 | 0.02945 | 5.89 |
| Style-B | 0.01061 | 0.00860 | 221.90 | 0.00984 | 0.00798 | 1120.09 | 0.01566 | 0.01227 | 5.90 |

This consistency across sampling densities provides strong evidence that `E-NES` effectively captures the underlying travel-time function for arbitrary point pairs throughout the domain. We attribute this capability to two key architectural decisions in our approach. First, the grid-free architecture allows the model to operate on continuous spatial coordinates rather than discretized grid positions. Second, our training methodology leverages physics-informed neural network (PINN) principles rather than relying on numerical solver supervision as implemented in Mei et al. [2024]. This physics-based learning approach enables `E-NES` to internalize the governing eikonal equation, resulting in a more generalizable representation of travel-time fields that remains accurate across varying evaluation protocols.

## E.3 Ablation Study: Choice of Embedding for Computing Invariants

To better understand the impact of architectural choices on model performance, we conducted an ablation study examining different embedding functions for computing invariants in our cross-attention module. Table 5 compares three embedding approaches: a standard MLP, a polynomial embedding (degree 3), and Random Fourier Features (RFF, our chosen approach).

The results demonstrate that RFF embeddings provide the best balance of accuracy and computational efficiency. While the polynomial embedding achieves comparable relative error (RE) and relative mean absolute error (RMAE) to RFF, it incurs a prohibitively high computational cost, with fit and inference time approximately 6 times slower. The standard MLP embedding, though computationally efficient, shows degraded performance across both metrics.

These findings justify our choice of RFF with learnable frequencies (as discussed in Section D.2), which combines superior accuracy with reasonable computational requirements.

Table 5: Ablation on the choice of embedding for computing invariants (CurveFault-B validation set).

| Embedding | RE ($\downarrow$) | RMAE ($\downarrow$) | Fit + Inf. Time (s) |
|---|---|---|---|
| MLP | 0.025 | 0.020 | 237.2 |
| Polynomial (deg. 3) | 0.024 | 0.019 | 1460.7 |
| RFF (Ours) | **0.021** | **0.016** | 255.8 |

## E.4 Ablation Study: Latent Conditioning Design

The design of the latent conditioning mechanism significantly impacts both model performance and computational efficiency. We investigated different configurations by varying the number of latent points ($N$) and their dimensionality ($d$), while maintaining a fixed total parameter budget of 288 (i.e., $N \times d = 288$). This ensures a fair comparison across different architectural choices.

Table 6 presents results on the CurveFault-B validation set. The configuration with 9 latent points of dimension 32 provides the optimal tradeoff between accuracy and computational cost, achieving the lowest error metrics (RE = 0.021, RMAE = 0.016) with reasonable inference time (255.8 seconds).

Table 6: Ablation on latent conditioning design (CurveFault-B validation set). Total latent budget fixed at 288 parameters.

| # Latents ($N$) | Latent Dim. ($d$) | RE ($\downarrow$) | RMAE ($\downarrow$) | Fit + Inf. Time (s) |
|---|---|---|---|---|
| 1 | 288 | 0.078 | 0.066 | 58.5 |
| 4 | 72 | 0.047 | 0.038 | 139.3 |
| 9 | 32 | **0.021** | **0.016** | 255.8 |
| 16 | 18 | 0.039 | 0.031 | 501.7 |

Notably, using a single high-dimensional latent vector ($1 \times 288$) results in significantly degraded performance, suggesting that spatial distribution of latent conditioning is crucial for capturing the complexity of the velocity field. Conversely, using too many low-dimensional latents ($16 \times 18$) increases computational cost without improving accuracy, likely due to insufficient expressiveness of the context vector per latent point. The selected configuration ($9 \times 32$) thus represents an effective compromise that provides both spatial coverage and sufficient representational capacity per latent point.

These results highlight the importance of carefully balancing the number and dimensionality of latent conditioning variables, and demonstrate that our chosen architecture is well-tuned for the problem at hand.

## E.5 Ablation Study: Impact of Autodecoding Fitting Epochs

This section presents a systematic analysis of how the number of autodecoding fitting epochs affects model performance and computational efficiency. We evaluate the E-NES model on the 2D-OpenFWI datasets across both grid configurations described in Section E.2, measuring performance in terms of Relative Error while tracking computational costs. Additionally, we provide a comparative analysis between the standard autodecoding approach and our meta-learning methodology.

Figures 7 and 8 illustrate the relationship between fitting time, number of epochs, and model performance across all datasets. Our analysis reveals that most datasets reach optimal solution convergence at approximately 400 autodecoding fitting epochs. Performance improvements beyond this threshold exhibit diminishing returns relative to the additional computational investment required. Notably, approximately 100 autodecoding epochs represents an effective compromise between computational efficiency and performance quality.

The meta-learning approach demonstrates remarkable efficiency advantages. With negligible fitting times compared to standard autodecoding, meta-learning achieves performance levels comparable to 50-100 epochs of autodecoding for the FlatVel-A/B and CurveVel-A/B datasets. This represents

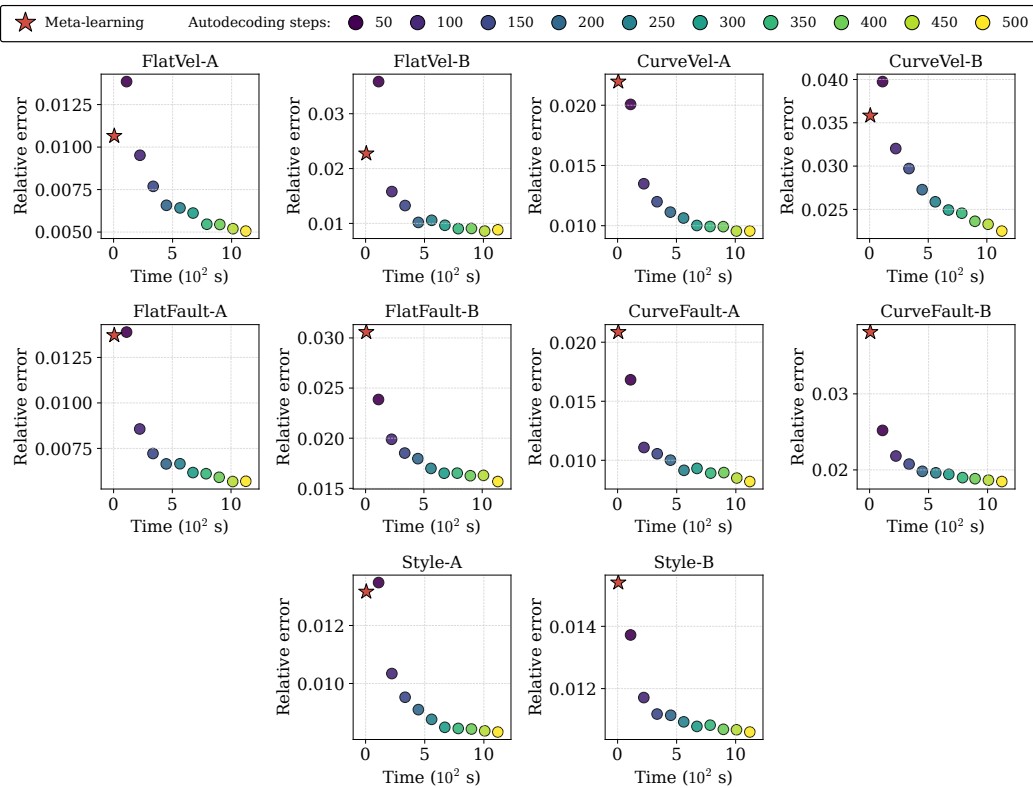

Figure 7: Comparative analysis of Relative Error versus total fitting time for all 100 velocity fields in the 4-source configuration across all datasets. Circular markers represent autodecoding performance at varying epoch counts (color-coded from 50 to 500 epochs), while the star marker indicates meta-learning performance.

a substantial reduction in computational requirements while maintaining acceptable performance characteristics.

These findings suggest that practitioners can optimize computational resource allocation by selecting the appropriate training approach based on their specific performance requirements and computational constraints. For applications where rapid deployment is critical, meta-learning offers significant advantages, while applications demanding maximum accuracy may benefit from extended autodecoding training, with the optimal epoch count determined by performance saturation points identified in our analysis.

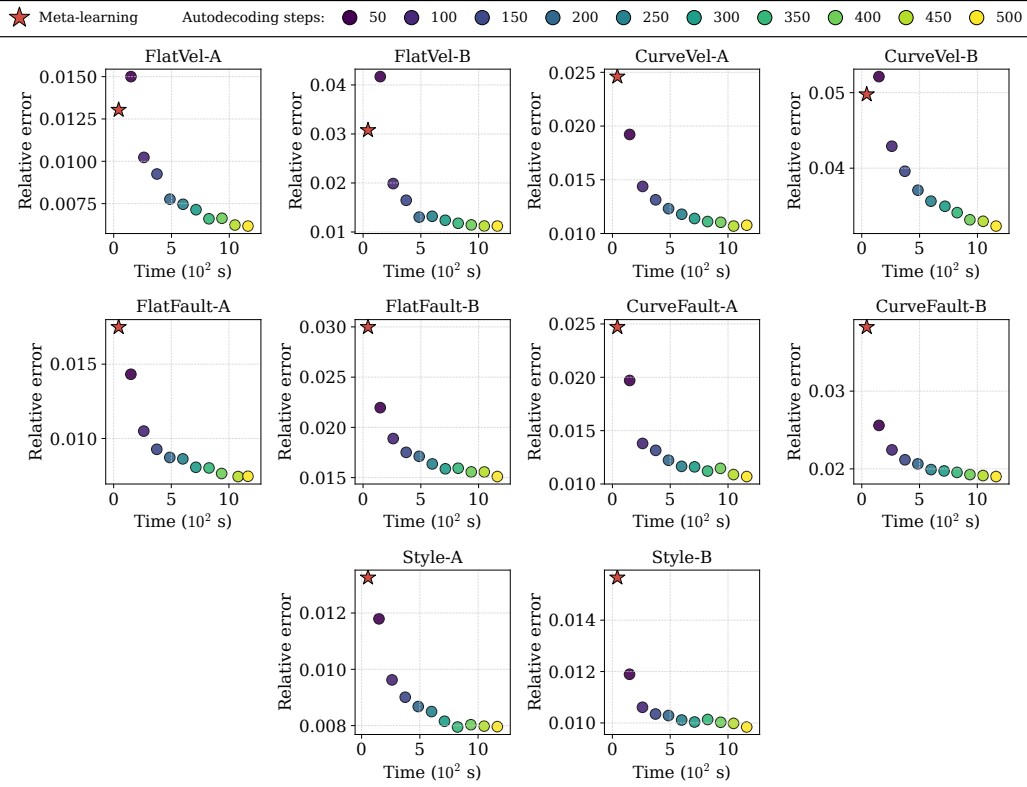

Figure 8: Comparative analysis of Relative Error versus total fitting time for all 100 velocity fields in the dense 14×14 source grid configuration. Circular markers represent autodecoding performance at varying epoch counts (color-coded from 50 to 500 epochs), while the star marker indicates meta-learning performance.

## E.6 Qualitative Analysis of Travel-Time Predictions

This section provides a visual assessment of the E–NES model's performance as quantitatively reported in Table 1. For each dataset in the 2D-OpenFWI benchmark, we present a representative velocity field alongside the corresponding ground-truth and predicted travel-time surfaces. Additionally, we visualize the spatial distribution of relative error to facilitate the identification of regions where prediction accuracy varies.

Figures 9 and 10 demonstrate the E–NES model's capacity to accurately reconstruct travel-time functions across diverse geological scenarios. Particularly noteworthy is the model's performance on the challenging CurveFault-A/B and Style-A/B datasets, where the travel-time functions exhibit complex wavefront behaviors including caustic singularities—regions. At these caustics, seismic-ray trajectories intersect each other, forming singularity zones where gradients are discontinuous.

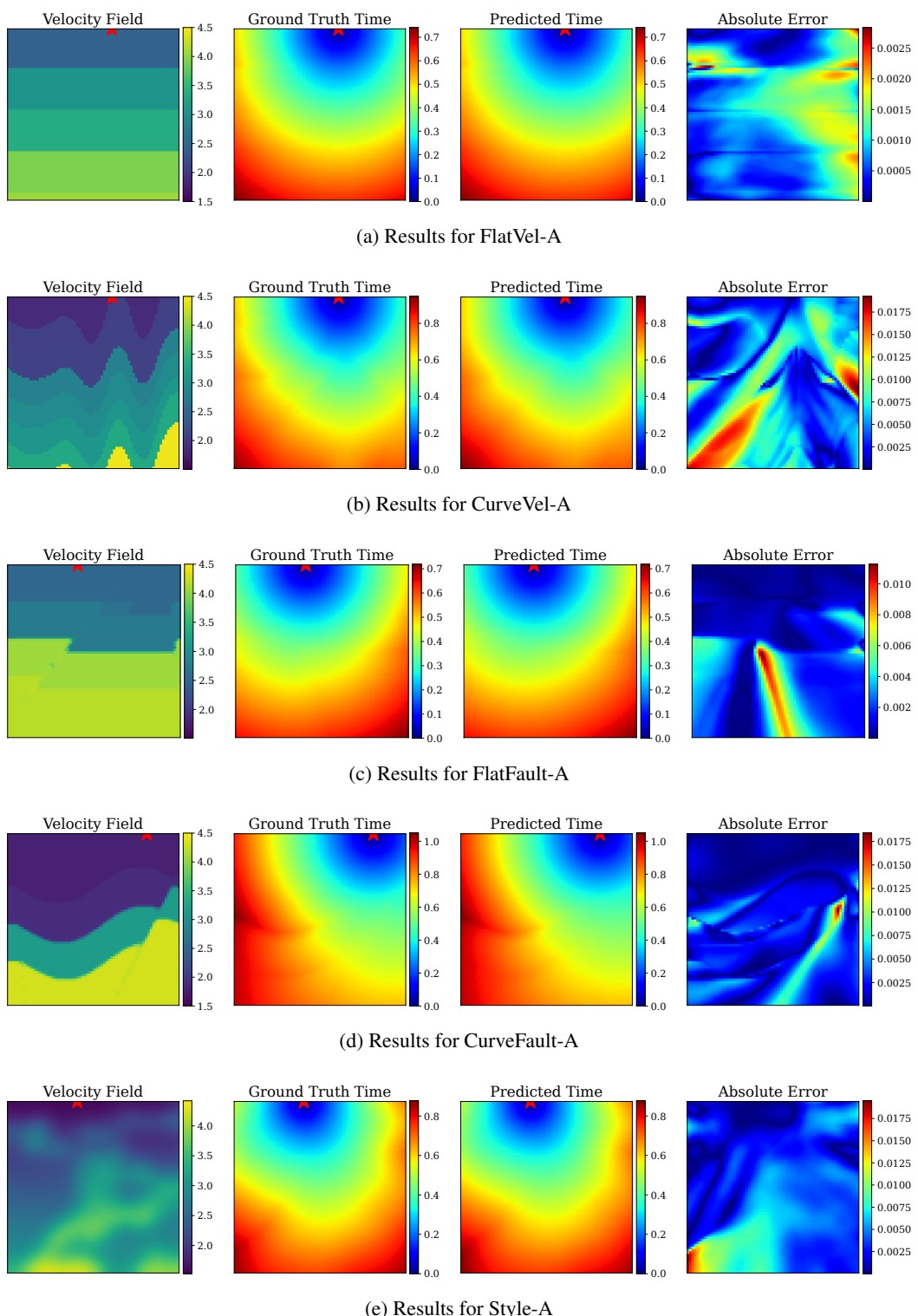

Figure 9: Comparative visualization of E-NES predicted travel-times against reference solutions for OpenFWI type B datasets. Each panel displays the velocity field (left), ground-truth travel-time surface (center-left), E-NES predicted travel-time surface (center-right), and relative error distribution (right). The red star denotes the source point location from which wavefronts propagate.

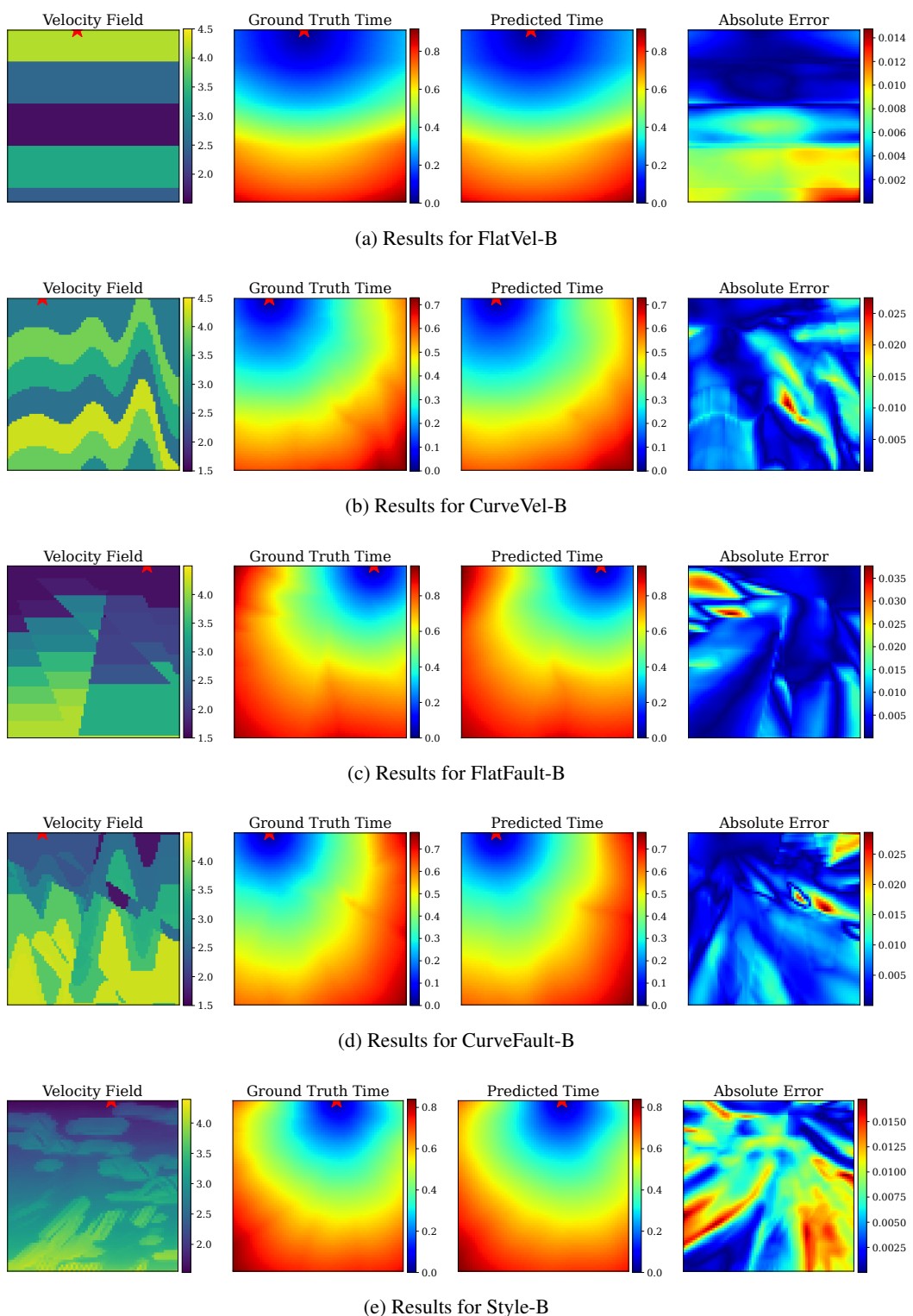

Figure 10: Comparative visualization of E-NES predicted travel-times against reference solutions for OpenFWI type A datasets. Each panel displays the velocity field (left), ground-truth travel-time surface (center-left), E-NES predicted travel-time surface (center-right), and relative error distribution (right). The red star denotes the source point location from which the wavefronts propagate.

### E.7 Empirical Steerability Test

In this section, we empirically validate the geometric inductive bias incorporated into our Eikonal Solver through a two-part test. First, we verify that $T_\theta(s, r; z_l)$ correctly models the solution of the Eikonal equation for the velocity field $v_l$. Second, we test the equivariance property: for any $g \in SE(2)$, we examine whether $T_\theta(s, r; g \cdot z_l)$ solves the Eikonal equation with the transformed velocity field $\mu(g, v_l)$, satisfying $\| \operatorname{grad}_s T(s, r; g \cdot z_l) \|_{\mathcal{G}}^{-1} = \mu(g, v_l)(s)$.

For the Special Euclidean group, Corollary 4.1 establishes that $\mu(g, v_l)(s) = v_l(g^{-1} \cdot s)$, which enables a direct geometric interpretation and visual verification of the steerability property. Figure 11 demonstrates this behavior empirically: when we rotate the latent conditioning point cloud, the resulting travel-time field exhibits a corresponding rotation, and the gradient norm yields a rotated version of the original velocity field. The same predictable equivariant behavior is observed under y-axis translation, confirming that our model successfully captures the desired geometric structure.

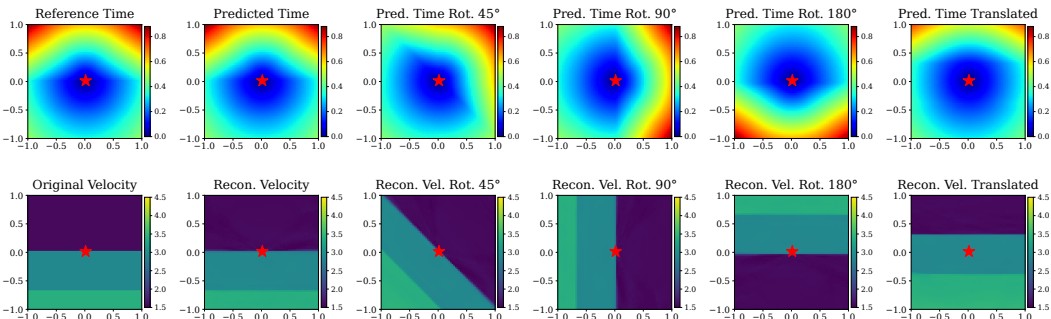

Figure 11: Steerability test demonstrating $SE(2)$ equivariance on the FlatVel-A dataset. When the latent conditioning point cloud is transformed by elements of $SE(2)$ (rotations and translations), the predicted travel-time field and recovered velocity field undergo corresponding geometric transformations, confirming the model's equivariant behavior. The red star denotes the source point from which wavefronts propagate.

## F    Extended Discussion

### F.1    Improved Memory Scalability

In this section, we expand our analysis to consider the relative merits of conditional versus unconditional neural field approaches for solving the eikonal equation. We intentionally excluded comparisons against unconditional neural field-based eikonal equation solvers from our main experimental results, as these comparisons require additional context to interpret appropriately.

Unconditional neural fields typically outperform conditional neural fields in both solution accuracy and inference speed for individual problem instances. This performance gap stems from the fundamental nature of the task: training an unconditional neural field essentially constitutes an overfitting problem, where even a modest MLP architecture can achieve high accuracy on a single velocity field. In contrast, conditional neural fields must generalize across multiple instances, effectively learning a mapping from conditioning variables to solution fields rather than memorizing a single solution.

However, this performance advantage diminishes significantly when considering parameter scalability across multiple velocity fields. The unconditional approach (e.g., a standard Neural Eikonal Solver) requires approximately 17,558 parameters per velocity field [Grubas et al., 2023], as it necessitates training an entirely new network for each travel-time solution. In contrast, our E-NES approach maintains a fixed baseline of 648,965 parameters for the core network $\tau_\theta$, with each additional velocity field requiring only 315 parameters for the conditioning variables (in the $\mathcal{Z} = SE(2) \times \mathbb{R}^{32}$ case). A simple calculation reveals that E-NES becomes more parameter-efficient than the unconditional approach when handling more than 38 velocity fields—a threshold easily surpassed in practical applications.

The memory scaling properties of E-NES offer additional advantages beyond parameter efficiency. The conditioning variables effectively serve as a quantization method for the travel-time function, significantly reducing memory requirements. For example, the 14×14×70×70 grid configuration used in Table 4 would require storing 960,400 floating-point values per velocity field using traditional methods. With E-NES, we need only 315 parameters per velocity field regardless of the grid resolution. This resolution-invariant property becomes particularly valuable as problem dimensions increase, offering orders of magnitude improvements in memory efficiency for high-resolution applications.

These scalability advantages highlight the complementary nature of conditional and unconditional approaches. While unconditional neural fields excel at individual problem instances where maximum accuracy is required, conditional architectures like E-NES provide substantially better scalability for applications involving multiple velocity fields or varying resolution requirements.

## F.2 Applicability Beyond Seismic Travel-Time Modeling

While our manuscript focuses primarily on the Eikonal equation in the context of seismic travel-time modeling, the proposed generalization of Equivariant Neural Fields (ENFs) extends far beyond this specific application.

**Eikonal Equation Beyond Seismic Imaging.** Even within the realm of Eikonal solvers, there exist several diverse and impactful application domains. For example, in *robotics*, the Eikonal equation is used for optimal path planning, where travel-time functions are defined over the robot's configuration space [Ni and Qureshi, 2023]. In *image segmentation*, the Eikonal equation can be employed to compute geodesic distances, with the velocity field derived from classical edge detectors, thereby transforming the segmentation task into a minimal-path extraction problem [Chen and Cohen, 2019]. Although our experiments focus on the OpenFWI benchmark—currently the only publicly available dataset for learning-based Eikonal solvers—our methods are directly applicable to these broader scenarios, as discussed in Section 1.

**Equivariant Neural Fields in Reinforcement Learning.** Beyond Eikonal solvers, the proposed framework is applicable in *continuous-control reinforcement learning* (RL). Specifically, the travel-time function $T(s, r)$ obtained by solving the Eikonal equation can be interpreted as an optimal value function:

$$v^*(s, r) = T(s, r) = \inf_{\gamma(0)=s,\, \gamma(1)=r} \int_0^1 \frac{\|\dot{\gamma}(t)\|}{v(t)} \, dt,$$

where the state space is $\mathcal{M} \times \mathcal{M}$, the action space corresponds to $T\mathcal{M}$, and the optimal policy is the geodesic path $\gamma$. The environment is deterministic, with transitions $(\gamma(\varepsilon), r)$ for small $\varepsilon > 0$, and terminal state $(r, r)$.

In general, RL problems with continuous state and action spaces, the value function $v$ and action-value function $q$ can naturally be modeled as neural fields. In *multi-task RL* [Yu et al., 2020], these functions are conditioned on task embeddings. Our formulation of Conditional Neural Fields is well-suited to this, as it accommodates both continuous coordinate inputs and a conditioning signal. Moreover, when the state or action spaces admit a Lie group symmetry [Wang et al., 2022], Equivariant Neural Fields offer inductive biases that promote better generalization and sample efficiency. Importantly, our proposed extension enables modeling such value functions in cases involving multiple input coordinates, which are common in practice.

**Beyond Reinforcement Learning.** The applicability of our method extends well beyond reinforcement learning. Consider, for instance, a collection of signals $f_1, \ldots, f_n$, where each $f_i : \mathcal{M}_i \to \mathbb{R}^d$ is defined over a potentially distinct Riemannian manifold $\mathcal{M}_i$. In many scientific and machine learning applications—including computational biology, multimodal sensor fusion, and medical imaging—it is of interest to model a *similarity metric* of the form

$$\text{sim}(f_1, \ldots, f_n)[p_1, \ldots, p_n],$$

which compares the values or features of these signals at specific points $(p_1, \ldots, p_n) \in \mathcal{M}_1 \times \cdots \times \mathcal{M}_n$. Traditional ENFs are limited to modeling functions defined over a single manifold input. However, our proposed extension allows us to define and learn such similarity metrics equivariantly with respect to group actions on the product manifold $\mathcal{M}_1 \times \cdots \times \mathcal{M}_n$.

### F.3    Additional Discussion on Performance Comparison against FC-DeepONet

As shown in Table 1, our method outperforms the FC-DeepONet baseline on 7 out of 10 OpenFWI datasets, with performance on FlatFault-A comparable. For the two datasets where our method underperforms (FlatVel-A/B), this can be attributed to two key factors:

**Training signal difference:**    FC-DeepONet is trained using the "ground-truth" travel time fields obtained from a numerical solver (FMM), rather than learning from the Eikonal equation directly. As discussed in Section 1, the quality of FMM solutions improves with finer domain discretization. In the FlatVel-A/B datasets, the velocity profiles have low spatial frequency (see Figures 8 and 9), making the FMM-derived travel times particularly accurate and advantageous as training targets. In contrast, our method learns from PDE constraints rather than supervised travel times. This makes our approach more broadly applicable, but also potentially disadvantaged in cases where numerical solvers already provide highly accurate approximations. Notably, this performance advantage for FC-DeepONet diminishes in more complex scenarios involving higher-frequency profiles or more intricate Riemannian manifolds, where numerical solvers like FMM may perform less reliably.

**Inductive bias from domain discretization:**    As discussed in Section 2, FC-DeepONet requires a discretized domain to produce conditional latents through its CNN encoder. While this introduces constraints on generalizability (e.g., limited applicability on manifolds with multiple charts or in tasks like geodesic backtracking), it can act as a strong inductive bias in low-frequency settings. The FlatVel-A/B datasets exemplify such settings, where this discretization bias likely aids FC-DeepONet's performance.

