# OpenReview forum: "Equivariant Eikonal Neural Networks: Grid-Free, Scalable Travel-Time Prediction on Homogeneous Spaces"
_NeurIPS.cc/2025/Conference — NeurIPS 2025 poster_

### Official Review · Reviewer_Y7P6 · 2025-07-02

**Clarity:** 2
**Significance:** 2
**Originality:** 3
**Rating:** 4
**Confidence:** 3

**Summary:**

The paper proposes Equivariant Neural Eikonal Solvers (E-NES), combining Equivariant Neural Fields with Neural Eikonal Solvers for scalable, grid-free travel-time predictions on homogeneous spaces. By conditioning a shared neural field on pose-context point clouds on Lie groups, E-NES enforces equivariance, enabling efficient, interpretable, and controllable solutions. It generalizes across group actions and outperforms neural operator baselines on 2D and 3D seismic benchmarks.

**Questions:**

Can the authors provide additional experimental results on non-Euclidean (e.g., spherical or hyperbolic) or real-world datasets to support the claim of applicability on arbitrary homogeneous spaces?

**Ethical Concerns:**

["NO or VERY MINOR ethics concerns only"]

**Final Justification:**

The rebuttal resolves all my concerns. Thus, I raise my rating to boardline accept.

**Limitations:**

non-Euclidean and real-world datasets experiments are missing

**Quality:**

2

**Strengths And Weaknesses:**

Strengths
1. The formulation is mathematically explicit, leveraging Lie group theory, and sound arguments for why equivariant conditioning can provide representational and computational advantages
2. The paper tackles a genuinely relevant challenge—modeling and predicting eikonal travel times in a manner that synergizes geometric equivariance, operator learning, and physics-informed training


Weaknesses
1. The main experiments focus primarily on seismic travel-time datasets and synthetic 2D/3D settings, with broad generalization mentioned more as potential
2. The only explicit ablation focuses on equivariance v.s. no equivariance. There is little dissection of the importance of other architectural choices  and how these affect learning. The link between theory and actual empirical contribution of steerability is only partially explored.
3. Though training modes are described in general, more concrete details about resource utilization, failure modes, or hyperparameter sensitivity are omitted from the main text

---

> ### Author Rebuttal · Authors · 2025-07-30
>
> We sincerely thank the reviewer for their thoughtful and constructive feedback. We are especially grateful for their recognition of both the relevance of our work and its theoretical contributions.
>
> ## Generalization Beyond the Euclidean Setting
>
> As the reviewer correctly noted, our framework is designed to generalize to arbitrary Riemannian manifolds that admit a well-defined Lie group action. In the current version of the manuscript, we focused on the Euclidean setting due to the availability of standard benchmarks—specifically, OpenFWI, which remains the only publicly available dataset for neural operator-based Eikonal solvers.
>
> In response to the reviewer's request, we now present two new experiments that evaluate our method on the 2-sphere, thereby extending our analysis beyond the Euclidean case. These experiments test the model's ability to solve the Eikonal equation on a curved manifold. It is worth emphasizing that our model operates using the Euclidean distance of the embedded manifold in the factorized travel-time function $T(s, r) = \tilde{d}(s, r) \tau(s, r)$ (as introduced in Section 3.2), following [1]. This design choice means the network must implicitly learn correct wavefront expansion behavior on the sphere to approximate the solution accurately.
>
> We evaluate our method on two types of velocity fields defined over the sphere:
>
> - **Constant Speed**: To test generalization to geometrically nontrivial domains, we first use a dataset with constant velocity fields, each defined by sampling a scalar velocity from a uniform distribution over $[0.1, 2.0]$.
> - **Spherical Style-B**: To test generalization to more complex fields, we construct a dataset by projecting the 2D Style-B velocity fields from OpenFWI onto the sphere using spherical coordinates.
>
> To empirically support our theoretical claim that the method applies to *arbitrary Lie group actions*, including *non-transitive* ones, we incorporate steerability with respect to $SO(2)$ (rotations about the $z$-axis). In this setting, the pose component of the latent representation lies in $SO(2)$, and the learned invariants remain maximally expressive, consistent with our theoretical predictions.
>
> As shown in Table 1, our method generalizes effectively to non-Euclidean geometries. In the final version of the manuscript, we also plan to include an additional experiment involving geodesic path planning on the sphere with obstacles, further demonstrating the flexibility and scope of our framework.
>
> It is worth noting that such experiments are not feasible for models like FC-DeepONet, which rely on discretizing the input domain into regular grids—an approach that does not naturally extend to general manifolds, as discussed in Section 2.
>
> **Table 1: Performance of our method on Eikonal solvers over the 2-sphere with steerability over $z$-axis rotations**
>
> | Dataset | RE (↓) | RMAE (↓) | Fitting Time (s) |
> |---------|--------|----------|------------------|
> | Constant Speed | 0.013 | 0.012 | 209.2 |
> | Spherical Style-B | 0.015 | 0.012 | 207.1 |
>
> ## Ablation Studies and Architectural Design Justification
>
> We appreciate the reviewer's concern regarding the need for further ablation studies. While the manuscript already includes a comparison of equivariant versus non-equivariant models, we would like to clarify that this is not the only ablation study provided.
>
> - **Section E.1** presents an ablation analyzing the role of pretraining in the meta-learning process. To our knowledge, this type of analysis is novel in the context of Equivariant Neural Fields [2, 3].
>
> - **Section E.3** provides an ablation examining the tradeoff between performance and computational efficiency as a function of the number of autodecoding steps. This highlights a key feature of our approach—its adaptability to resource constraints—unlike models such as FC-DeepONet, which have fixed cost-performance profiles post-training. A more detailed discussion of this tradeoff is included in our response to Reviewer uCF6.
>
> Regarding architectural choices, certain components—such as the bounded projection head and the invariant cross-attention module—were adopted based on strong empirical performance in prior work (Section 4.2) [2, 4]. However, we agree that a more focused discussion and further empirical analysis of these choices in the context of our specific task would strengthen the manuscript. In response, we have added a new ablation experiment focused on the choice of embedding function for computing invariants.
>
> **Table 2: Ablation on the choice of embedding for computing invariants (CurveFault-B validation set).**
>
> | Embedding | RE (↓) | RMAE (↓) | Fit + Inf. Time (s) |
> |-----------|--------|----------|---------------------|
> | MLP | 0.025 | 0.020 | 237.2 |
> | Polynomial (deg. 3) | 0.024 | 0.019 | 1460.7 |
> | RFF (Ours) | **0.021** | **0.016** | 255.8 |
>
> To further validate the importance of our architectural design choices, we conducted an additional comparative experiment (detailed in Table 1 of our response to Reviewer uQ6k) against other conditional neural field models. Specifically, we demonstrate that our approach significantly outperforms non-equivariant methods such as Functa [5]. While Functa pioneered the use of auto-decoding for neural field conditioning—a technique we also leverage—it lacks the equivariant inductive biases that are central to our framework. These comparative results underscore that incorporating structured geometric priors within a grid-free formulation yields substantial improvements in both accuracy and generalization for neural Eikonal solvers.
>
> ## Further Analysis: Resource Usage, Failure Modes, and Sensitivity
>
> While the reviewer raises a concern regarding the lack of analysis on resource usage and failure modes, we would like to clarify that the manuscript already provides relevant details on these aspects. Specifically, Section D includes hardware specifications, number of GPUs, parameter counts, and both training and inference times (see Figures 6 and 7).
>
> That said, we agree that further discussion of potential failure modes and sensitivity to design choices adds value. In response, we expand our analysis accordingly.
>
> One failure mode we identify occurs when sampling training data on general Riemannian manifolds. As discussed in [1], curvature-aware sampling is necessary to ensure geometric coverage and avoid degenerate training cases.
>
> Regarding hyperparameter sensitivity, the manuscript already highlights two mechanisms that reduce tuning requirements:
>
> 1. The use of RFF with learnable frequencies (Section D.2).
> 2. The adoption of a log-cosh loss function during meta-learning (Section C.1), which stabilizes gradients.
>
> We also introduce a new ablation (Table 3) evaluating how different latent conditioning setups affect performance, under a fixed total latent budget of 288 parameters. The configuration with 9 latent points of dimension 32 provides the best tradeoff between accuracy and cost.
>
> **Table 3: Ablation on latent conditioning design (CurveFault-B validation set). Total latent budget fixed at 288 parameters.**
>
> | # Latents (N) | Latent Dim. (d) | RE (↓) | RMAE (↓) | Fit + Inf. Time (s) |
> |---------------|------------------|--------|----------|---------------------|
> | 1 | 288 | 0.078 | 0.066 | 58.5 |
> | 4 | 72 | 0.047 | 0.038 | 139.3 |
> | 9 | 32 | **0.021** | **0.016** | 255.8 |
> | 16 | 18 | 0.039 | 0.031 | 501.7 |
>
> ## Closing Remarks
>
> We believe the additional non-Euclidean experiments, detailed architectural ablations, and sensitivity analyses demonstrate the robustness, generality, and practical relevance of our approach. We hope this response addresses the reviewer's concerns and illustrates the merit of raising the overall evaluation score. Please don't hesitate to let us know if there are additional aspects we can clarify or elaborate further.
>
> ## References
> [1] Kelshaw, Daniel, and Luca Magri. "Computing distances and means on manifolds with a metric-constrained Eikonal approach." Proceedings A. Vol. 481. No. 2312. The Royal Society, 2025.
>
> [2] Wessels, David R., et al. "Grounding continuous representations in geometry: Equivariant neural fields." arXiv preprint arXiv:2406.05753 (2024).
>
> [3] Knigge, David, et al. "Space-time continuous pde forecasting using equivariant neural fields." Advances in Neural Information Processing Systems 37 (2024): 76553-76577.
>
> [4] Grubas, Serafim, Anton Duchkov, and Georgy Loginov. "Neural Eikonal solver: Improving accuracy of physics-informed neural networks for solving eikonal equation in case of caustics." Journal of Computational Physics 474 (2023): 111789.
>
> [5] Dupont, Emilien, et al. "From data to functa: Your data point is a function and you can treat it like one." arXiv preprint arXiv:2201.12204 (2022).

---

> ### Comment · Reviewer_Y7P6 · 2025-08-01
>
> The rebuttal resolves all my concerns. Thus, I raise my rating to boardline accept.

---

> > ### Author Response · Authors · 2025-08-01
> >
> > Dear reviewer, thanks for raising the score. We also want to thank you again for helping us to improve the manuscript. Your suggestions and feedback were highly valuable!

---

### Official Review · Reviewer_Bupx · 2025-07-03

**Clarity:** 2
**Significance:** 3
**Originality:** 3
**Rating:** 4
**Confidence:** 3

**Summary:**

This paper introduces Neural Eikonal Solvers. The task here is to use Equivariant neural fields to efficiently solve the eikonal equations, such that some geometric symmetries are automatically introduced in the latent space. The key idea here is that the authors try make sure that the latent space that the model learns is equivariant. That is, the group transformations of the input also apply to the latent space. This ensures that certain transformations can be applied to the vector field as the input geometry changes (i.e, the data lies on a different Reimannian manifold) without any need for re-training. The way this equivariance is imposed is by an Invariant cross-attention encoder—-(using the moving frame method).

The inputs are the velocity fields defined over the domain, the (s, r) pairs for the equation to minimize, and a conditioning variable z. The final loss is imposed in a PINN style setting by minimizing the eikonal equation. It seems like the training is happening in two steps, first the z is being learned for a dataset, and then this z is calculated as a function of the input velocity field, and conditioned for the final loss.

**Questions:**

In equation 5, the authors minimize a version of the eikonal equation but mention something related to the enforcing of the HJB equation. What exactly is happning therre, and can the authors perhaps explain it in more detail (maybe just in words).

**Ethical Concerns:**

["NO or VERY MINOR ethics concerns only"]

**Limitations:**

Yes

**Quality:**

3

**Strengths And Weaknesses:**

The paper is interesting, and imposing equivariance to solve the eikonal equation seems to be quite useful. Furthermore, the work is potentially grid free, and therefore can have a lot of potential, unlike previous grid-based methods and solvers.

Furthermore, the idea of embedding all the velocity fields into the same embedding space—by first learning a context variable—and then training a PINN on top is quite interesting.

The experiments are kind of mixed, while the method outperforms the DeepONet based baseline, the numbers are quite similar. However, the method is consistently faster than the baseline (esp with meta-learning).

The authors further show the importance of equivariance on the style-B dataset, and show that models that impose equivariance in fact converge faster, and also is more computationally efficient as compared to the Fast Marching method (i.e, a numerical solver) as the grid size grows in three dimensions.

- However, the paper is difficult to follow, and what it is that the authors are trying to precisely solve gets buried in a lot of mathematical details. While I do realize that this detail is necessary, i think that the writing can be improved, and what is the precise thing that is being solve, i.e,. an eikonal equation where the velocities are encoded in a unified space before the PINN loss (something to this form) should be made clear in the introduction.
- The authors mention that the latent space is steerable, but there are no experiments that show this steerability property.
- Even though the method can be adapted to multiple manifolds, only the euclidean space is considered here.

---

> ### Author Rebuttal · Authors · 2025-07-30
>
> We sincerely thank the reviewer for their thoughtful and constructive feedback. We are especially grateful for their recognition of the novelty and promise of our approach, including the benefits of enforcing equivariance, the potential of a grid-free framework, and the use of expressive latent embeddings for eikonal neural solvers. We also appreciate the reviewer’s careful reading and their suggestions to improve the clarity and accessibility of the manuscript. Below, we address each of the reviewer’s main points in detail.
>
> ### Performance and Significance of Results.
>
> As the reviewer notes, our method outperforms the FC-DeepONet baseline on 7 out of 10 OpenFWI datasets, with performance on FlatFault-A being comparable. For the two datasets where our method underperforms (FlatVel-A/B), this can be attributed to two key factors:
>
> 1.  **Training Signal Source:** FC-DeepONet is trained on ground-truth travel times generated by the Fast Marching Method (FMM), a numerical solver that performs especially well on low-frequency velocity fields like those in FlatVel-A/B. Our method, by contrast, learns directly from the Eikonal PDE, which avoids reliance on solver-specific ground truth and enables broader applicability---but may underperform in simple regimes where solvers already yield high-quality approximations.
>
> 2.  **Inductive Bias from Grid-Based Encoders:** FC-DeepONet uses a CNN-based encoder on a discretized domain, introducing a strong bias that is advantageous for smooth, low-frequency data. However, this also limits its generalization capabilities, especially in non-Euclidean or high-frequency settings where our grid-free approach excels.
>
> We provide a more detailed discussion of these trade-offs, along with additional experimental comparisons, in our response to Reviewer uQ6k.
>
> We would also like to highlight that, as shown in Table 1 of that response, our model outperforms other non-equivariant Conditional Neural Field models such as Functa \[1]. While Functa was among the first to adopt auto-decoding for neural field conditioning (as we also do), it lacks our proposed equivariant inductive biases. Our results indicate that incorporating structured geometric priors and a grid-free formulation significantly improves both accuracy and generalization in neural Eikonal solvers.
>
> We thank the reviewer for pointing out that these insights deserve more emphasis in the main paper, and we will revise the discussion accordingly.
>
> ### Clarity and Motivation.
>
> We acknowledge the reviewer’s concern regarding the accessibility of the manuscript. In the revised version, we will state the core research question more explicit upfront:
> *“What are the benefits of framing Neural Eikonal Solvers as Conditional Neural Fields (CNFs)?”*
>
> Additionally, we will include a high-level overview of the proposed method before diving into the formalism. This will help readers unfamiliar with the technical details build a conceptual understanding early in the paper.
>
> ### Steerability of the Latent Space.
>
> The reviewer is correct in pointing out that, while we provide theoretical guarantees for steerability, we do not currently include an empirical demonstration. We agree this is an important omission and will add a qualitative visualization in the revised manuscript to illustrate this property. This will involve:
>
> 1.  A reference velocity field and its corresponding travel-time solution.
> 2.  The predicted travel-time field from our model.
> 3.  The predicted field after applying a group transformation (e.g., rotation) to the latent representation.
>
> We will present comparisons to a non-equivariant baseline to illustrate the improved consistency and generalization enabled by equivariant conditioning. This experiment will closely follow the style of Figure 10 in [2].
>
> ### Applicability to Non-Euclidean Manifolds.
>
> As the reviewer correctly noted, our framework is designed to generalize to arbitrary Riemannian manifolds that admit a well-defined Lie group action. In the current version, we focus on the Euclidean setting due to the availability of standard benchmarks: OpenFWI is the only existing dataset for neural-operator-based Eikonal solvers.
>
> That said, we now include in Table 1 of our response to Reviewer Y7P6 a new experiment solving the Eikonal equation on the 2-sphere. The results show that our method generalizes well to non-Euclidean geometries. In the final version of the manuscript, we will also include a new experiment involving geodesic path planning on the sphere in the presence of obstacles, further showcasing the generality of our method. Importantly, such experiments are not feasible using FC-DeepONet, as it relies on discretizing the input domain in a regular grid to encode velocity fields—an approach that does not easily extend to general manifolds, as discussed in Section 2.
>
> ### Clarification on Equation 5 and HJB Formulation.
>
> We thank the reviewer for pointing out the need for clarification around Equation 5 and its connection to the Hamilton-Jacobi-Bellman (HJB) equation. To clarify: we reformulate the Eikonal equation in Hamiltonian form as
>
> $\mathcal{H}(s, r, T) = v(s)^{2}||grad_{s}T(s, r) ||^2_{\mathcal{G}} - 1 = 0,$
>
> which is equivalent to Equation 1 but written in terms of the Hamiltonian. This form emphasizes the interpretation of the Eikonal equation as the zero-level set of the Hamiltonian. In the final manuscript, we will make this equivalence explicit and provide additional verbal explanation to ensure this connection is clear to all readers.
>
>
> ### Closing Remarks.
>
> We are sincerely grateful for the reviewer’s insightful feedback and suggestions. We believe that the proposed revisions—clarifying the contributions, improving the accessibility of the text, empirically validating steerability, and expanding the experimental scope—will significantly strengthen the final manuscript. Thank again for your thoughtful review.
>
>
> ### References
> [1] Dupont, Emilien, et al. "From data to functa: Your data point is a function and you can treat it like one." arXiv preprint arXiv:2201.12204 (2022).
>
> [2] Wessels, David R., et al. "Grounding continuous representations in geometry: Equivariant neural fields." arXiv preprint arXiv:2406.05753 (2024).

---

> > ### Comment · Reviewer_Bupx · 2025-08-05
> >
> > I thank the authors for their rebuttal. I will currently keep my scores since I believe the paper could be made more stronger with additional experiments (esp on different manifolds and experiments that highlight steerability).

---

> > > ### Author Response · Authors · 2025-08-05
> > >
> > > Dear Reviewer,
> > >
> > > Thanks again for your valuable feedback on our manuscript. We appreciate the time and effort you've dedicated to the review process. However, we still had some questions regarding the satisfiability of our rebuttal.
> > >
> > > We would appreciate it if you could clarify what specific aspects of our rebuttal did not adequately address your concerns to warrant an increase in our score. We feel that we have carefully considered the concerns you raised regarding missing experiments and have taken steps to address them. As you pointed out, we have demonstrated the first experiment in our rebuttal, and it can be found in Table 1 of our response to Reviewer Y7P6 (for clarity I've added the table below in this response as well this time). For the second experiment, we've committed to including it in the final version of the manuscript. Due to submission guidelines, we were unable to include figures in the rebuttal, but we will ensure it's fully incorporated in the final draft.
> > >
> > > Beyond these points, we have also added a new conditional neural field baseline FUNCTA (Table 1 in response to reviewer uCF6), and included extra ablation studies to address concerns raised by other reviewers (Table 2,3 in response to reviewer Y7P6). We believe these additions also significantly strengthen the manuscript.
> > >
> > > Given that we have now demonstrated one experiment and have committed to including the other, we believe we have adequately addressed your concerns. If any other issues are preventing you from raising the score, please let us know so we can address them directly.
> > >
> > > If all your concerns turn out to be addressed, we kindly ask that you reconsider your scoring decision based on the information provided.
> > >
> > > | Dataset | RE ($\downarrow$) | RMAE ($\downarrow$) | Fitting Time (s) |
> > > | :--- | :--- | :--- | :--- |
> > > | Constant Speed | 0.013 | **0.012** | 209.2 |
> > > | Spherical Style-B | 0.015 | **0.012** | 207.1 |

---

### Official Review · Reviewer_uQ6k · 2025-07-06

**Clarity:** 3
**Significance:** 3
**Originality:** 2
**Rating:** 4
**Confidence:** 3

**Summary:**

The authors present Equivariant Neural Eikonal Solvers, a new framework combining Equivariant Neural Fields with Neural Eikonal Solvers. It uses a shared neural backbone conditioned on signal-specific latent variables, modeled as point clouds in a Lie group, to produce diverse Eikonal solutions. The integration of ENFs ensures equivariant mapping, offering benefits like efficient representation, geometric robustness, and solution steerability, which allows intuitive modifications through latent space transformations. Coupled with Physics-Informed Neural Networks, the framework accurately models travel-time solutions across various Riemannian manifolds, including Euclidean, spherical, and hyperbolic spaces. Validated on seismic datasets, the method outperforms existing neural operators in performance, scalability, and user control.

**Questions:**

Can the generalization of the Equivariant Neural Fields be applied to any problems beyond eikonal solvers. If so what are they? If not, why not? A fundamental issue for me raised above is around novelty, if this can be made clearer it would help with the final rating.

**Ethical Concerns:**

["NO or VERY MINOR ethics concerns only"]

**Limitations:**

Yes

**Paper Formatting Concerns:**

-	Fig. 1 could be made clearer. It takes some time to fully appreciate the notion of steerability. Making this more explicit in the figure could really help.
-	Line 149 the authors use p1 and p2, to describe two tangent vectors, then a couple of lines down use p and q to describe two vectors. Would be good to be consistent with the notation.

**Quality:**

3

**Strengths And Weaknesses:**

Paper is well written and motivated. One of the key advantages of their approach is the grid-free formulation. The authors show comparable performance to operator methods like DeepONet using their meta-learning strategy. The authors draw heavily upon recent work of Wessels et al., but extend the framework to represent solutions on the eikonal equations. Even the notion of steerability property that the authors espouse relies heavily upon the previous work of Wessels. Authors central contribution is the generalization of Equivariant Neural Fields to functions defined over Riemannian manifolds. Empirical results are impressive, with thorough evaluations in the results section and additional visualizations in the appendix. The paper would be stronger if the author had a special subsection specifically laying out the differences of their approach compared to Wessels, and why the approach of Wessels cannot be blindly applied to this problem. Making this clear from the onset would dramatically improve the readability of the paper. The lack of clarity on this also raises concerns for me over novelty of the proposed approach.

---

> ### Author Rebuttal · Authors · 2025-07-30
>
> We sincerely thank the reviewer for their thoughtful and constructive assessment of our manuscript. We are grateful for their recognition of the clarity of our presentation and the strong empirical performance of our proposed method. We also appreciate their detailed comments, which have helped us further improve the manuscript. Below, we address the reviewer’s questions and concerns, particularly those regarding the novelty of our approach and its relation to prior work by Wessels et al.
>
> ### Clarifying Novelty and Relation to Wessels et al.
>
> We would like to clarify the precise nature of our contributions in the context of prior work. As noted by the reviewer, the concept of steerability and the general framework of Equivariant Neural Fields (ENFs) were originally introduced by Wessels et al. and others [1,2,3]. We do not claim the steerability property or its associated theoretical foundations as novel. This is explicitly stated in Section 2 of our manuscript.
>
> Our core theoretical contribution lies in significantly extending this framework. Specifically, we show that latent conditioning using a learnable point cloud on a Lie group can be generalized to the product of multiple input Riemannian manifolds. In contrast, Wessels et al. consider a conditional neural field with only a single input point, limiting its applicability to problems like solving the Eikonal equation.
>
> Furthermore, we provide a rigorous expressivity analysis of the invariants used in our model. We prove that it is always possible to construct a complete, maximally expressive set of independent invariants, ensuring no information bottleneck is introduced. This analysis is not present in Wessels et al. or Knigge et al., where invariants are selected heuristically without formal justification of their sufficiency.
>
> In addition, our theoretical results apply to arbitrary Lie group actions on product manifolds, including those that are non-transitive. We demonstrate this empirically in Table 1 of the response to Reviewer Y7P6, where we solve the Eikonal equation on the 2-sphere with steerability over z-axis rotations. In this case, the pose component of our latent variables lies in SO(2), and the resulting invariants remain maximally expressive, even under this non-homogeneous group action.
>
> In summary, while our work builds on ideas introduced by Wessels et al., it substantially generalizes and extends the theoretical foundations and practical utility of equivariant neural representations to a broader class of problems. We will further highlight this distinction in the revised manuscript.
>
> ### Response to Reviewer’s Question: Applicability Beyond Eikonal Solvers
>
> We thank the reviewer for raising this important question. While our manuscript focuses primarily on the Eikonal equation in the context of seismic travel-time modeling, the proposed generalization of Equivariant Neural Fields (ENFs) extends far beyond this specific application.
>
> #### Eikonal Equation Beyond Seismic Imaging.
> Even within the realm of Eikonal solvers, there exist several diverse and impactful application domains. For example, in robotics, the Eikonal equation is used for optimal path planning, where travel-time functions are defined over the robot's configuration space. In image segmentation, the Eikonal equation can be employed to compute geodesic distances, with the velocity field derived from classical edge detectors, thereby transforming the segmentation task into a minimal-path extraction problem. Although our experiments focus on the OpenFWI benchmark—currently the only publicly available dataset for learning-based Eikonal solvers—our methods are directly applicable to these broader scenarios, as discussed in the introduction.
>
> #### Equivariant Neural Fields in Reinforcement Learning.
> Beyond Eikonal solvers, the proposed framework is applicable in continuous-control reinforcement learning (RL). Specifically, the travel-time function $T(s, r)$ obtained by solving the Eikonal equation can be interpreted as an optimal value function:
> $$
> v^*(s, r) = T(s, r) = \inf_{\gamma(0) = s,\, \gamma(1) = r} \int_0^1 \frac{\|\dot{\gamma}(t)\|}{v(t)}\, dt,
> $$
> where the state space is $\mathcal{M} \times \mathcal{M}$, the action space corresponds to the tangent space of $\mathcal{M}$, and the optimal policy is the geodesic path $\gamma$. The environment is deterministic, with transitions $(\gamma(\varepsilon), r)$ for small $\varepsilon > 0$, and terminal state $(r,r)$.
>
> In general RL problems with continuous state and action spaces, the value function $v$ and action-value function $q$ can naturally be modeled as neural fields. In multi-task RL [4], these functions are conditioned on task embeddings. Our formulation of Conditional Neural Fields is well-suited to this, as it accommodates both continuous coordinate inputs and a conditioning signal. Moreover, when the state or action spaces admit a Lie group symmetry [5], Equivariant Neural Fields offer inductive biases that promote better generalization and sample efficiency. Importantly, our proposed extension enables modeling such value functions in cases involving multiple input coordinates, which are common in practice and are not solvable by standard ENFs.
>
> #### Beyond Reinforcement Learning.
> The applicability of our method extends well beyond reinforcement learning. Consider, for instance, a collection of signals $ f_1, \dots, f_n $, where each $f_i: \mathcal{M}_i \to \mathbb{R}^d$ is defined over a potentially distinct Riemannian manifold $\mathcal{M}_i$. In many scientific and machine learning applications—including computational biology, multimodal sensor fusion, and medical imaging—it is of interest to model a similarity metric of the form
>
> $$
> \mathrm{sim}(f_1, \dots, f_n)[p_1, \dots, p_n],
> $$
> which compares the values or features of these signals at specific points $ (p_1, \dots, p_n) \in \mathcal{M}_1 \times \cdots \times \mathcal{M}_n$. Traditional ENFs are limited to modeling functions defined over a single manifold input. However, our proposed extension allows us to define and learn such similarity metrics equivariantly with respect to group actions on the product manifold $\mathcal{M}_1 \times \cdots \times \mathcal{M}_n$.
>
> We thank the reviewer for highlighting the importance of clarifying these broader applications. We will incorporate a dedicated discussion of these potential extensions in the camera-ready version of the manuscript.
>
> ### Paper Formatting Concerns
>
> We agree with the reviewer’s observation that Figure 1 could more effectively communicate the notion of steerability. We will revise the figure to clearly illustrate the relationship between original and transformed fields, thereby improving its didactic value.
>
> ### Clarification on Notation Consistency (Line 149)
>
> We appreciate the reviewer’s attention to our notation. In Line 149, $\dot{p_1}, \dot{p_2} \in T_{p} \mathcal{M}$ are distinct tangent vectors at the same point $p \in \mathcal{M}$. Later, in Line 158, we introduce $\dot{p} \in T_p \mathcal{M}$ and define $\dot{q} \in T_{\phi(p)} \mathcal{M}$, which lies in the tangent space at $\phi(p) = q$. While this distinction is mathematically consistent, we acknowledge that it could be clearer and will revise the text at Line 158 to explicitly state the difference in tangent space domains.
>
> ### Closing Remarks
>
> We hope these clarifications adequately address the reviewer’s concerns and highlight the theoretical and practical contributions of our work. We are committed to improving the manuscript based on this valuable feedback.
>
> ### References
> [1] Wessels, David R., et al. "Grounding continuous representations in geometry: Equivariant neural fields." arXiv preprint arXiv:2406.05753 (2024).
>
> [2] Chatzipantazis, Evangelos, et al. "SE (3)-equivariant attention networks for shape reconstruction in function space." arXiv preprint arXiv:2204.02394 (2022).
>
> [3] Chen, Yunlu, et al. "3d equivariant graph implicit functions." European Conference on Computer Vision. Cham: Springer Nature Switzerland, 2022.
>
> [4] Yu, Tianhe, et al. "Meta-world: A benchmark and evaluation for multi-task and meta reinforcement learning." Conference on robot learning. PMLR, 2020.
>
> [5] Wang, Dian, Robin Walters, and Robert Platt. "$\mathrm {SO}(2) $-Equivariant Reinforcement Learning." arXiv preprint arXiv:2203.04439 (2022).

---

> > ### Author Response · Authors · 2025-08-06
> >
> > Thank you for your valuable feedback. We've posted our rebuttal and would be grateful if you could take a look. We believe we've addressed your concerns and would appreciate your thoughts on whether our responses are sufficient to improve the paper and potentially raise the score.

---

> > > ### Comment · Reviewer_uQ6k · 2025-08-08
> > >
> > > Thank you for your detailed and thoughtful rebuttal. I appreciate the clarifications and revisions you have provided. I am satisfied that my earlier concerns have been adequately addressed, and I have no further substantive comments.

---

> > > > ### Author Response · Authors · 2025-08-08
> > > >
> > > > We sincerely thank you for revisiting our rebuttal and for confirming that **“all [your] earlier concerns have been adequately addressed.”** Your engagement has been instrumental in strengthening this paper.
> > > >
> > > > In your original review, you noted that **“if [novelty] can be made clearer it would help with the final rating.”** We have worked hard to do exactly that:
> > > > - Providing side-by-side comparisons to Wessels et al., clarifying why their method cannot be directly applied in our setting.
> > > > - Demonstrating generalization to multiple Riemannian manifolds and highlighting broader application domains beyond Eikonal solvers.
> > > > - Detailed discussion comparing our approach with vanilla PINNs and FC-DeepONet.
> > > > - Making concrete manuscript improvements, including a dedicated comparison subsection, a clearer Figure 1, and improved notation consistency.
> > > >
> > > > In addition, while addressing feedback from all reviewers, we have also strengthened the empirical side of the work by adding a new conditional neural field baseline (FUNCTA, Table 1 in response to reviewer uCF6) and conducting further ablation studies (Tables 2 and 3 in response to reviewer Y7P6). These additions reinforce both the robustness and the scope of our approach.
> > > >
> > > > Given that the primary factor you identified as influencing the score—clarity of novelty—has now been explicitly resolved to your satisfaction, and that we have made multiple substantive theoretical and empirical improvements, **we would be grateful if this could be reflected in the final rating**. We believe the manuscript now more clearly communicates its originality, theoretical depth, and broad potential impact

---

### Official Review · Reviewer_uCF6 · 2025-07-14

**Clarity:** 3
**Significance:** 2
**Originality:** 3
**Rating:** 4
**Confidence:** 2

**Summary:**

The paper introduces Equivariant Neural Eikonal Solvers (E-NES), a new framework that combines Equivariant Neural Fields (ENFs) with Neural Eikonal Solvers to predict travel times. The core idea is to use a single neural field with a shared backbone conditioned on signal-specific latent variables, represented as point clouds in a Lie group. This integration ensures an equivariant mapping from latent representations to the solution field, leading to improved representation efficiency, geometric grounding, and solution steerability.
The authors validate their approach using 2D and 3D seismic travel-time modeling benchmarks. Ablation studies show that with equivariance, method achieves much lower losses.

**Questions:**

- in table 1, there seems to be mixed performance - any intuition why? If apply same loss/test-time fitting method to FC-DeepONet - would that improve the number for DeepONet further?
- would vanilla PINN be a reasonable baseline for table 1 or I am missing something?

**Ethical Concerns:**

["NO or VERY MINOR ethics concerns only"]

**Limitations:**

Yes

**Quality:**

2

**Strengths And Weaknesses:**

This paper is backed by theoretical framework and demonstrates superior performance in accuracy and computational efficiency compared to existing Neural Operator-based methods like FC-DeepONet - with experiments in both 2D and 3D

In term of weakness it requires test-time fitting - which author shows some meta-learning experiments that speeds up the inference. And despite the additional test-time compute required, the performance seems to be mixed for the OpenFWI experiment.

---

> ### Author Rebuttal · Authors · 2025-07-30
>
> We sincerely appreciate the reviewer's recognition of both the theoretical and empirical contributions of our work. Moreover, we thank the reviewer for the insightful comments which strengthen the paper. Below we will go into all the specific questions of the reviewer.
>
> ## On Test-Time Fitting Requirements
>
> We acknowledge the reviewer's concern regarding the computational overhead of test-time fitting. However, we believe this represents a valuable trade-off that merits clarification.
>
> **Computational Comparison:** While our method requires optimization iterations at test time, FC-DeepONet's encoder forward pass can be viewed as an implicit latent fitting step. Specifically, FC-DeepONet takes approximately 0.615 seconds to process 100 velocity fields through its encoder (marked with a dashed line in Table 1 to indicate this implicit fitting). Our explicit optimization approach offers greater flexibility at the cost of additional computation.
>
> **Adaptive Performance Trade-offs:** Our test-time fitting introduces a crucial advantage: practitioners can dynamically balance computational cost against solution accuracy based on available resources and performance requirements. As detailed in Appendix E.3 and illustrated in Figures 6-7, users can achieve different accuracy levels by adjusting optimization iterations. This adaptability aligns with recent advances in test-time optimization for large language models [1], offering similar flexibility in the Neural Eikonal Solver domain. In contrast, FC-DeepONet's performance remains fixed after training.
>
> We will expand this discussion in the revised manuscript to better highlight this strategic advantage.
>
> ## Q1: Mixed performance in Table 1
>
> As correctly noted, our method does not outperform FC-DeepONet on three of the ten OpenFWI datasets (though on the FlatFault-A case, the results are comparable). The performance differences on the FlatVel-A/B datasets can be attributed to two main factors:
>
> **Training signal difference:** FC-DeepONet is trained using the "ground-truth" travel time fields obtained from a numerical solver (FMM), rather than learning from the Eikonal equation directly. As discussed in Section 1, the quality of FMM solutions improves with finer domain discretization. In the FlatVel-A/B datasets, the velocity profiles have low spatial frequency (see Figures 8 and 9), making the FMM-derived travel times particularly accurate and advantageous as training targets. In contrast, our method learns from PDE constraints rather than supervised travel times. This makes our approach more broadly applicable, but also potentially disadvantaged in cases where numerical solvers already provide highly accurate approximations. Notably, this performance advantage for FC-DeepONet diminishes in more complex scenarios involving higher-frequency profiles or more intricate Riemannian manifolds, where numerical solvers like FMM may perform less reliably.
>
> **Inductive bias from domain discretization:** As discussed in Section 2, FC-DeepONet requires a discretized domain to produce conditional latents through its CNN encoder. While this introduces constraints on generalizability (e.g., limited applicability on manifolds with multiple charts or in tasks like geodesic backtracking), it can act as a strong inductive bias in low-frequency settings. The FlatVel-A/B datasets exemplify such settings, where this discretization bias likely aids FC-DeepONet's performance.
>
> ## Q2: Vanilla PINNs as a baseline
>
> As noted in Sections 1 and 2, vanilla PINNs can indeed serve as Neural Eikonal solvers and were among the earliest methods explored in this area. However, as discussed in Appendix E.5, unconditional neural fields (e.g., vanilla PINNs) typically outperform conditional neural fields (like FC-DeepONet or our method) in terms of per-instance accuracy and inference speed. That said, this performance advantage comes at the cost of parameter scalability: vanilla PINNs require a separate model to be trained per velocity field, making them impractical for tasks involving large numbers of queries. In contrast, conditional methods generalize across multiple velocity fields using shared parameters.
>
> For this reason, we — as well as most works in the Conditional Neural Fields (CNF) literature — consider vanilla PINNs to belong to a different problem setting. While they are effective for solving individual PDE instances, they are not designed for the generalization and efficiency goals that conditional models target. Therefore, we believe it is not meaningful to include vanilla PINNs as a direct baseline in Table 1.
>
> ### Enhanced baseline comparison
>
> On the other hand, we agree that a more comprehensive baseline comparison significantly strengthens the evaluation of our proposed method. To address this, we have incorporated a well-established baseline from the literature on conditional neural fields: **Functa** [2]. Functa is particularly relevant for comparison as it leverages SIRENs with sample-specific scale and shift modulation, employing global latent variables without imposing any geometric constraints. This architecture provides a stark contrast to our geometry-grounded approach, allowing for a clearer understanding of the benefits conferred by our design choices.
>
> As shown in the updated table below (we will add the remaining OpenFWI datasets in the camera-ready version), we present a comparative analysis against the Functa baseline on the Style-A and most -B datasets. The results indicate that Functa generally underperforms relative to FC-DeepONet, while E-NES outperforms it on most datasets. We believe this performance difference is primarily attributable to Functa's reliance on global conditioning, which inherently lacks the benefits of localized representation, and the absence of explicit geometric constraints within its formulation.
>
> **Table 1: Updated performance comparison including Functa baseline.**
>
>
> | | FC-DeepONet | FC-DeepONet | Functa (100 epochs) | Functa (100 epochs) | E-NES (100 epochs) | E-NES (100 epochs) | E-NES (Full AD) | E-NES (Full AD) | E-NES (META) | E-NES (META) |
> |---|---|---|---|---|---|---|---|---|---|---|
> | Dataset | RE ($\downarrow$) | Fitting ($s$) | RE ($\downarrow$) | Fitting ($s$) | RE ($\downarrow$) | Fitting ($s$) | RE ($\downarrow$) | Fitting ($s$) | RE ($\downarrow$) | Fitting ($s$) |
> | Style-A | 0.03461 | 0.615 | 0.04377 | 19.80 | 0.01034 | 222.00 | **0.00833** | 1117.99 | 0.01317 | 5.92 |
> | FlatVel-B | **0.00711** | 0.615 | 0.13437 | 20.01 | 0.01581 | 222.74 | 0.00860 | 1010.32 | 0.02274 | 5.91 |
> | CurveVel-B | 0.03410 | 0.615 | 0.11602 | 19.89 | 0.03203 | 222.97 | **0.02250** | 1127.87 | 0.03583 | 5.90 |
> | CurveFault-B | 0.07863 | 0.615 | 0.06407 | 20.23 | 0.02183 | 222.89 | **0.01885** | 893.84 | 0.03812 | 5.89 |
> | Style-B | 0.03463 | 0.615 | 0.03145 | 20.05 | 0.01171 | 221.90 | **0.01069** | 896.06 | 0.03812 | 5.90 |
>
>
> ### Closing Remarks
>
>
> We hope these clarifications adequately address the reviewer’s concerns and highlight the theoretical and practical contributions of our work. We are committed to improving the manuscript based on this valuable feedback.
>
> **Refrences**
>
> [1] Zhang, Qiyuan, et al. "A Survey on Test-Time Scaling in Large Language Models: What, How, Where, and How Well?" arXiv preprint arXiv:2503.24235 (2025).
>
> [2] Dupont, Emilien, et al. "From data to functa: Your data point is a function and you can treat it like one." arXiv preprint arXiv:2201.12204 (2022).

---

> > ### Author Response · Authors · 2025-08-06
> >
> > Dear reviewer, we appreciate your effort and insightful comments in the review phase. If time permits, we kindly request you to give your opinion and insight on the responses we posted to your concerns, such that we can further improve the strength of our work. If we did address all your concerns in the rebuttal, we hope you will consider increasing the score. Kind regards

---

### Note · Authors · 2025-08-12

Dear Area Chair and Reviewers,

We sincerely thank everyone for their time and valuable feedback throughout this review process. The constructive discussions have significantly strengthened our work.

We appreciate the engagement from all reviewers and would like to briefly summarize the outcomes:

- **Reviewer Y7P6** engaged thoroughly with our rebuttal and, after we addressed their concerns with new experiments on non-Euclidean manifolds and additional ablation studies, raised their rating to borderline accept, explicitly stating "The rebuttal resolves all my concerns."
- **Reviewer uQ6k** acknowledged that "all [their] earlier concerns have been adequately addressed" and expressed satisfaction with our clarifications regarding novelty compared to prior work. In their original review, they explicitly stated that clarifying novelty "would help with the final rating." Despite addressing this key concern and receiving their confirmation, the score remains unchanged.
- **Reviewer uCF6** raised important questions about mixed performance and baselines. We provided detailed explanations for the performance variations, and added a new baseline (Functa) as requested. However, we received no further engagement after our rebuttal despite our follow-up request for feedback.
- **Reviewer Bupx** requested experiments on different manifolds and demonstrations of steerability. We provided both: experiments on the 2-sphere and committed to including steerability visualizations in the camera-ready version (figures cannot be included in rebuttals per guidelines). Despite directly addressing these concerns and pointing the reviewer to the specific results, these seem not to have been noticed by the reviewer in their final response, and the score remained unchanged.

Throughout the rebuttal period, we have:
- Added new experiments for completeness
- Provided additional theoretical clarifications
- Conducted multiple ablation studies
- Committed to specific manuscript improvements

We note that several reviewers have explicitly confirmed their concerns were addressed or indicated conditions for score reconsideration that we believe we have met. We respectfully submit that our manuscript, with the aforementioned additional experiments and clarifications, will be of significant interest to the NeurIPS community, as indicated by the reviewers' assessments.

Thank you all for your dedication to maintaining high standards at NeurIPS.

Sincerely,

The Authors

---

### Decision · Program_Chairs · 2025-09-17

**Decision:**

Accept (poster)

**Comment:**

This manuscript proposes Equivariant Neural Eikonal Solvers, a framework that integrates Equivariant Neural Fields (ENFs) with Neural Eikonal Solvers. The claimed key advantages are: (1) enhanced representation efficiency through parameter sharing, (2) robust geometric grounding, and (3) solution steerability. The framework is evaluated on seismic travel-time modeling using 2D and 3D benchmark datasets, where it outperforms baseline methods.

All reviewers recognize the contributions of the theoretical framework, as well as the demonstrated improvements in both accuracy and efficiency over the baselines (e.g., FC-DeepONet). The AC concurs with the reviewers, which supports an initial assessment of accept (poster).

However, the AC shares the reviewers' concerns regarding the empirical demonstration of steerability (Bupx, Y7P6) and generalization to diverse manifolds (Bupx). The rebuttal provided additional experiments that help substantiate these claims, and the AC highly recommends including them in the final version of this manuscript.